# Spatially separate production of hydrogen oxides and nitric oxide in lightning

Jena M. Jenkins[1], William H. Brune[1]

[1]Department of Meteorology and Atmospheric Science, Pennsylvania State University, University Park, PA, USA

*Correspondence to*: Jena M. Jenkins (jzj76@psu.edu)

**Abstract.** The atmosphere's most important oxidizer, the hydroxyl radical (OH), is generated in abundance by lightning, but the contribution of this electrically generated OH (LOH) to global OH oxidation needs to be better quantified. Part of the uncertainty in this contribution is due to the abundant nitric oxide (NO) also generated in lightning, which would rapidly remove the LOH before it can oxidize other pollutants in the atmosphere. However, atmospheric observations and a previous

laboratory study show extreme LOH coexists with extreme NO. The only way this $LHO_x$ can possibly survive is if LOH production is spatially separated from the NO production in lightning flashes and laboratory sparks. This hypothesis of spatially separate OH and NO production is further tested here in a series of laboratory experiments, where the OH decays were measured from spark discharges in air which had increasing amounts of NO added to it. The LOH decayed faster as more NO was added to the air, indicating that the LOH was reacting with the added NO, and not the spark NO. Thus, LOH from lightning

flashes is not immediately consumed by the electrically generated NO but is available to oxidize other pollutants in the atmosphere and contribute to global OH oxidation. Subsequent modelling of the laboratory data also supports the spatially separate production of LOH and NO and further suggests that substantial HONO may also be produced by sparks and lightning in the atmosphere.

## 1 Introduction

Lightning and other electrical discharges have been shown to directly generate extreme amounts of the atmosphere's primary oxidant, the hydroxyl radical (OH), and the closely related hydroperoxyl radical ($HO_2$) in field studies (Brune et al. 2021; Brune et al., 2022), laboratory studies (Jenkins et al. 2021; Ono and Oda, 2002), and modelling studies (Bhetanabhotla et al., 1985; Ripoll et al. 2014). The first reported field measurements of electrically generated OH and $HO_2$ (together called the hydrogen oxides or $HO_x$) were from the Deep Convective Clouds and Chemistry campaign in 2012, where as much as ~2 ppbv

of electrically generated $HO_x$ ($LHO_x$) was measured (Brune et al., 2021). Subsequent laboratory studies showed that both lightning and weaker cloud electrical discharges, called corona discharges, were generating the extreme amounts of $LHO_x$ and that $LHO_x$ was initially generated as equal amounts of LOH and $LHO_2$ (Jenkins et al., 2021). Based on these studies, lightning and corona discharges in thunderstorms are together estimated to account for as much as 2-16% of global OH. However,

narrowing down the uncertainty of this range will require more work. The frequency, duration, and location of corona

discharges is not well known, complicating attempts to estimate global OH production from these discharges.

In comparison, it is accepted that lightning flashes occur at a rate of 44 s$^{-1}$ globally (Christian et al., 2003), last <1 second (Rakov and Uman, 2006), and are mostly detected by satellites and lightning networks. The extreme amount of nitrogen oxide (NO) also generated in lightning makes estimating the impact of LHO$_x$ difficult, as theoretically this NO will rapidly remove

the extreme OH before it oxidizes other chemical species in the atmosphere, such as methane, carbon monoxide, sulfur dioxide, or other pollutants. However, evidence from a previous laboratory study shows that LHO$_x$ is not immediately destroyed by electrically generated NO (LNO). In Jenkins et al. (2021), laboratory sparks were generated inside a flow tube, and the subsequent LNO and LHO$_x$ formed from these discharges was measured. Hundreds of pptv of LHO$_x$ was observed to decay over hundreds of milliseconds, while simultaneously 1-2 ppmv of LNO was also measured. When these same measurements

of LHO$_x$ and LNO from the laboratory experiments were input into a photochemical box model, the Framework for 0-D Atmospheric Modelling (F0AM) (Wolfe et al., 2016) with the Master Chemical Mechanism v3.3.1 (Jenkin et al., 2015), the model predicted that LNO should have titrated all LHO$_x$ away in less than 10 ms, a small percent of the hundreds of milliseconds over which the LHO$_x$ decay was actually observed. It is unlikely that this discrepancy is due to some unimagined chemistry considering how well studied this chemistry is. Therefore, the only logical conclusion is that LHO$_x$ and LNO

generation are spatially separated for the spark, preventing their immediate reaction.

Spatially separate production is possible due to the structure of and different types of energy present in lightning flashes and sparks. At the center of a lightning flash is a ~1-2 cm diameter core (Rakov and Uman, 2006) with air temperatures exceeding 30,000K (Orville, 1968a). Surrounding this hot core is a weaker area of electrical discharge, called the corona sheath. The air

temperature in the corona sheath is near ambient, and the electrical discharges from the sheath extend radially several meters from the hot core (Rakov and Uman, 2006), so that the ratio of the volume of the corona sheath to the volume of the core is at least 10$^4$:1. Some of the radiation emitted by lightning flashes is in the ultraviolet (UV) range, composed of both broad spectrum and line emissions (Orville, 1968b), and including wavelengths <300 nm that are emitted from the sun but normally not present in the troposphere due to their absorption in the higher levels of the atmosphere by ozone. This UV radiation is generated by

both the hot core and the corona sheath. The reach of the UV radiation depends on the wavelength and scattering the radiation encounters but can be as much as tens of meters. Sparks are essentially a smaller scale version of lightning flashes, still composed of a hot core (though not as hot as lightning) surrounded by a weaker and cooler corona sheath and emitting UV radiation (though not as much as lightning).

The differences between the core and corona sheath lead to different chemistry occurring in each area. For example, the extremely high temperatures of the lightning flash or spark core are required to dissociate stable N$_2$ and make the extreme amounts of NO present in lightning flashes via the Zel'dovich Mechanism (Chameides et al., 1977):

$$O_2 \leftrightarrow O + O \qquad (\text{R1})$$
$$N_2 + O \leftrightarrow NO + N \qquad (\text{R2})$$
$$N + O_2 \leftrightarrow NO + O \qquad (\text{R3})$$

The air cools down rapidly after the lightning flash, removing the energy required for the reverse reactions to convert NO back to $N_2$ and $O_2$ faster than these reactions can occur. As a result, elevated NO remains after the lightning flash is completed. Conversely, without the high temperatures, the corona sheath makes several orders of magnitude less LNO (Rehbein and Cooray, 2001; Bhetanabhotla et al., 1985), so less than 1% of the spark NO is made outside the core. However, large amounts of OH, though not $HO_2$, are also made by combustion at the high temperatures of the core (Dyer and Crosley, 1982; Bhetanabhotla et al. 1985; Ripoll et al., 2014), while both OH and $HO_2$ are made through multiple pathways in the corona sheath. These pathways include, for example, OH-forming reactions like $electron + H_2O \rightarrow OH + H$ or $O^1D + H_2O \rightarrow 2OH$, reactions that form $HO_2$ like $H + O_2 + M \rightarrow HO_2 + M$, or UV radiation directly dissociating water vapor at wavelengths <200 nm, directly producing equal amounts of OH and $HO_2$:

$$H_2O + h\nu \rightarrow OH + H \qquad (\text{R4})$$
$$H + O_2 + M \rightarrow HO_2 + M \qquad (\text{R5})$$

In short, LNO production is contained in the very narrow hot core, while $HO_x$ production occurs in both the hot core and in a volume extending several meters the outside the hot core in the corona sheath. Thus, spatially separate $LHO_x$ and LNO production is possible.

To further test the hypothesis that $LHO_x$ and LNO production are spatially separated in spark discharges, we conducted a series of laboratory experiments in which the LOH and $LHO_2$ decays from spark discharges in air were measured with different amounts of background NO added into the air flow, from 0 ppbv up to 1000 ppbv of added NO. The decays from the laboratory experiments are also compared to decays calculated by F0AM with MCM to see if the model can successfully reproduce these decays. If $LHO_x$ decays faster as the background NO mixing ratio is increased, then $LHO_x$ is mostly or entirely reacting with background NO instead of spark LNO, confirming that $LHO_x$ and LNO generation is spatially separated in the spark. Otherwise, if the $LHO_x$ decays are unaffected by the amount of added NO, then $LHO_x$ is mostly or entirely reacting with spark LNO, $LHO_x$ and LNO are likely generated in the same location, and some unimagined chemistry is causing the discrepancy between model and measurement.

## 2 Methods

### 2.1 Laboratory Experimental Setup

The laboratory setup was nearly identical to the setup used in our previous LHO$_x$ studies (Jenkins et al. 2021; Jenkins and Brune, 2023). Purified and dried air, with an OH reactivity of ~0.35 s$^{-1}$ (Brune & Jenkins, 2024), was flowed through a bubbler to add a controlled amount of water vapor, then mixed with dry air that flowed down a quartz (previously Pyrex®) tube (50 mm OD x 46 mm ID x 105 cm) at 50 standard liters per minute (slpm), through spark discharges, and over to instruments for measuring OH and HO$_2$ (Ground-based Tropospheric Hydrogen Oxides Sensor [GTHOS; Faloona et al., 2004]), NO-NO$_2$-NO$_x$ (ECO PHYSICS nCLD 855Y), and O$_3$ (Kalnajs & Avallone, 2010). A solid-state Tesla coil (Eastern Voltage Research, Plasmasonic® 1.3) was used to generate the sparks across a 0.7 cm gap between tungsten wire electrodes (0.10 cm diameter) inside the flow tube. The sparks were generated in packets of 10 sparks, with ~75 ms between each spark in the packet, as signals from individual sparks were too narrow to consistently measure even at the 5 Hz sampling rate of GTHOS. The NO$_x$ analyzer collected data at a rate of 2 Hz and the O$_3$ analyzer collected data at a rate of 1 Hz. Each electrode was attached to a copper rod; one copper rod was attached via a copper wire cable to the output toroid of the Tesla coil, while the other was attached to an electrical ground. All discharges were generated using the same Tesla coil settings. Pressure (MKS Baratron® Type 222) was monitored ahead of the inlet for GTHOS and the Teflon tubing leading to the NO$_x$ and O$_3$ analyzers, temperature was measured both before air entered the flow tube (Vaisala HMT310) and as the air exited (thermistor), and the water vapor mixing ratio (Vaisala HMT310) was also measured before the air entered the flow tube. The air velocity was measured with an anemometer (TSI Inc., 8455-09) before running experiments, and the flow in the tube was previously determined to be laminar that is not fully developed (Jenkins et al., 2021). A short piece of Teflon tubing (1.3 cm diameter x 2.5 cm long) was placed on the GTHOS inlet, and the opening of the Teflon tube leading to the NO$_x$ and O$_3$ analyzers was positioned ~2 mm downstream of the GTHOS opening and facing into the short piece of Teflon tubing. This arrangement ensured that GTHOS and the NO$_x$ and O$_3$ analyzers all sampled from the same volume. The absolute uncertainty and limit of detection at the 68% confidence level was ±20% and ~1 pptv for the HO$_x$ measurements from GTHOS, ±10% and ~1–3 ppbv for the NO$_x$ measurements, and ±5% and ~20 ppbv for the O$_3$ measurements. A diagram of the laboratory setup is shown in Figure 1.

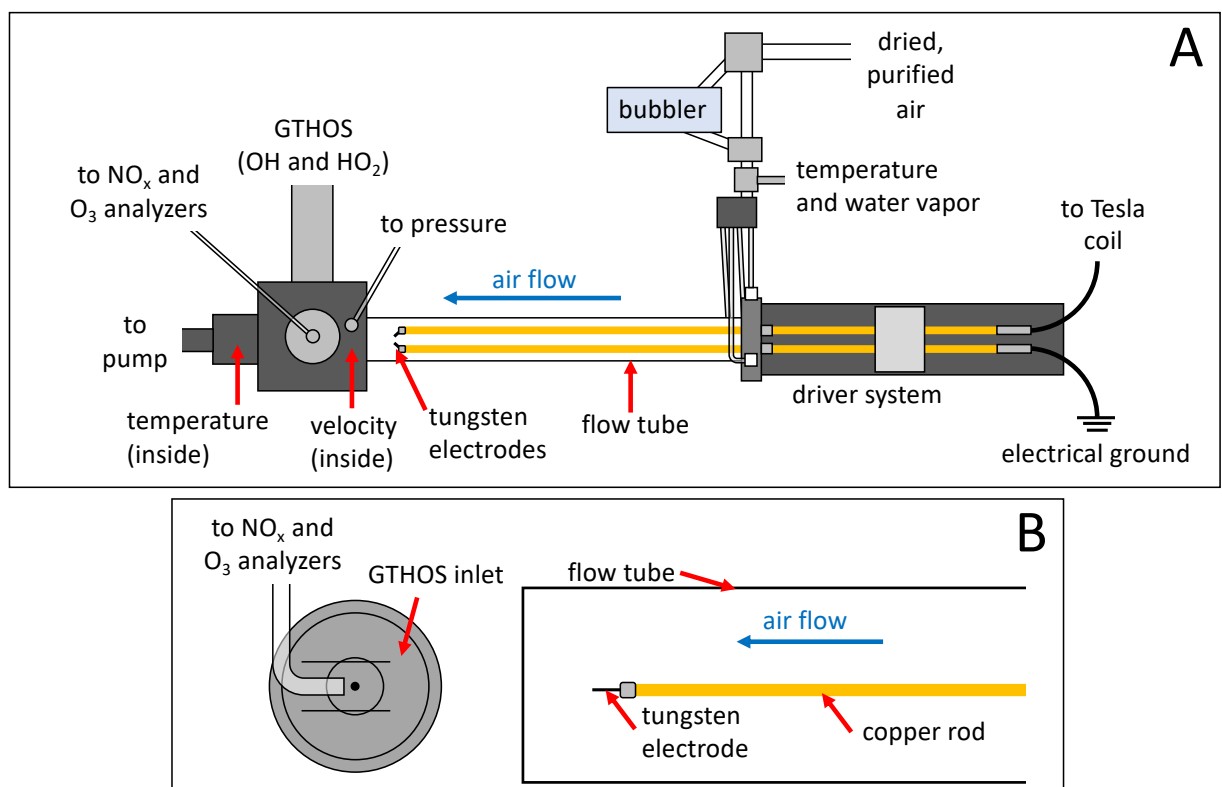

**Figure 1:** (A) Top-down diagram of the laboratory experimental setup showing the key components. (B) Side-view showing a close-up of the GTHOS inlet and Teflon line leading to the $NO_x$ and $O_3$ analyzers, which sample from the same volume as GTHOS in a 1.3-cm diameter tube place over the GTHOS inlet (shown as two horizontal lines), along with the relative positions of the flow tube, copper rod, and tungsten electrode. Neither (A) nor (B) is shown to scale.

As in real lightning, the core of the sparks is small relative to volume available for the corona sheath and ultraviolet radiation to occupy. Based on the visible light, the spark core is estimated to be ~1 mm in diameter across the 0.7 cm spark gap, so the core volume occupies ~0.006 $cm^3$. Assuming a similar ratio of corona sheath to core in the sparks that is present in lightning, then the spark corona sheath can occupy a volume as large as 55 $cm^3$ with a radius of 5 cm, although the 4.6 cm inner diameter of the flow tube will be the actual cut off point for the corona sheath. We can detect the UV radiation from the spark discharges with a spectrometer placed outside of the flow tube, so the UV radiation also travels well beyond the spark core. The air that the NO instrument samples through a 4 m long ¼" Teflon line is well mixed, indicating that the actual core NO is much higher than measured. However, GTHOS pulls 6 slpm and thus each 0.2 second measurement contains a volume of surrounding air hundreds of times larger than the ~0.006 $cm^3$ volume of core air. Because GTHOS is right at the exit of the flow tube and the flow is laminar and 50 slpm, molecular diffusion mixes the core air into a volume less than 1 $cm^3$ at the longest reaction time of ~0.5 s. Thus, GTHOS samples both core and sheath air, but they are spatially separated in the flow tube when sampled.

The experiments were conducted as follows. To capture the LHO$_x$ decay, the copper rods were moved by a driver system so discharges were generated in 5 different positions in the flow tube, over a total distance of 27.5 cm. In each position, four spark packets were generated, with 5 second spacing between each packet. For one of the four spark packets, the laser on GTHOS was switched to a wavelength slightly off the OH absorption wavelength to confirm the absence of electrical interference in the OH and HO$_2$ signals. By moving the discharge, the distance between the discharge and instrument inlets was changed, which also changed the time between the LHO$_x$ generation and measurement, producing the LHO$_x$ decay over time. The different amounts of added NO in the system were created by adding NO (Linde, 4.83 ppm) to the air flow before it entered the flow tube to create mixing ratios of 0, 50, 100, 250, 500, or 1000 ppbv (all within ± 6%). Because lightning can occur at any pressure in the troposphere, data were collected at pressures of 970 hPa, 770 hPa, 570 hPa, and 360 hPa (all within ± 2%) to cover most of the tropospheric pressures. Data were also collected at water vapor mixing ratios between 2000-2400 ppmv and temperatures between 289-294K.

Normally GTHOS uses two detection axes to simultaneously measure OH and HO$_2$, but only one detection axis was available when these experiments were conducted. To obtain both OH and HO$_2$ measurements for these experiments, OH was measured in a set of experiments, and total HO$_x$ was measured in another set of experiments conducted under the same conditions. The average OH measured at each position was subtracted from the total HO$_x$ generated at the same position and collected under the same conditions to determine the HO$_2$ generated.

## 2.2 Laboratory Data Processing

Each 10-spark discharge packet created a single spike in the OH, HO$_2$, NO, and NO$_x$ signals. Figure 2 shows the OH and NO signals from the spark packets over time for one experiment. No O$_3$ was detected in these experiments. These spikes were integrated over time to determine the total amount of chemical generated by the spark discharge. For the OH and HO$_2$ measurements, the peaks were about ~1.2 seconds wide and were integrated over 2.2 seconds, while the NO and NO$_x$ peaks were ~4.8 seconds wide and were also integrated over 4.8 seconds. From previous tests, only about 85% of the generated LNO$_x$ is sampled (Jenkins et al., 2021), so the LNO and LNO$_2$ results were corrected up 15% to account for the LNO$_x$ that is not sampled. OH and HO$_2$ have similar diffusion coefficients to NO$_x$ (Tang et al., 2014), so OH and HO$_2$ were also corrected up 15% to account for sampling. Additionally, the lifetime of NO$_x$ is long relative to the time it spends in the flow tube (hours vs <0.5 seconds, respectively), so any change in the NO$_x$ mixing ratio across the different positions was assumed to come from diffusion and not chemical loss. The average change in NO$_x$ over the different discharge positions in the flow tube is shown in Figure S1 for all four pressures tested. The LOH and LHO$_2$ measurements were also corrected up based on the NO$_x$ diffusion to account for diffusion losses.

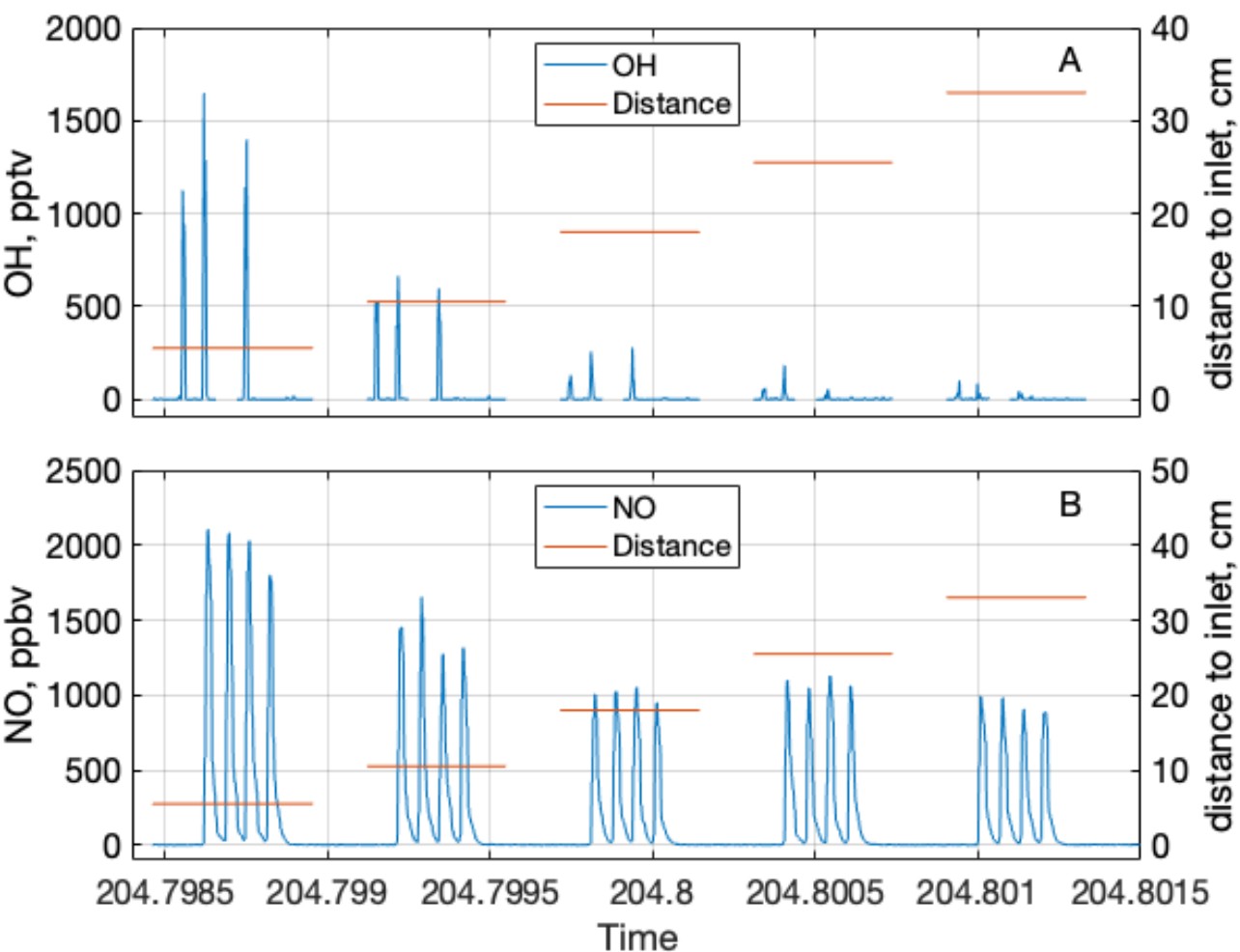

**Figure 2:** Change in OH (A) and NO (B) mixing ratios due to the spark discharges at each of the five discharge positions at 770 hPa and 0 ppbv of added NO. Each peak is from one spark packet containing 10 sparks. OH and NO mixing ratios are indicated by the blue lines and use the y-axes on the left side of their respective subplots, while the distances from the discharge to the GTHOS inlet and Teflon line leading to the $NO_x$ analyzer are indicated by the orange lines and use the y-axes on the right side.

Both the LOH and $LHO_2$ decays were fitted with equations assuming constant, first-order losses. These equations were extrapolated back to time-zero to determine the initial amount of these species generated in the discharge. In some experiments, the $HO_x$ decay was fast enough that the $HO_x$ data became too small and imprecise to use at farther discharge positions in the flow tube. If at least 3 positions had clear OH and $HO_2$ signals, the decay was included in the results; if only 2 positions or less

were available, the data were not used in the results, as there was not enough confidence in the extrapolated fit. Consequently, not all pressures have results for all the different amounts of added NO.

The initial $LNO_x$ formed in the discharges was taken as the $LNO_x$ in the position closest to the instrument inlets as it was least affected by diffusion. $NO_2$ made up <10% of total $NO_x$.


## 2.3 Model Setup

The modelling experiments were conducted using F0AM v3 with MCM 3.3.1 chemistry. The laboratory data were collected in 10 spark packets, but the chemical measurements were scaled down to single spark equivalents before inputting them into the model. The reason for scaling down is two-fold. First, even at the slowest speed in the flow tube, one spark will travel ~7

cm before the next one occurs, and previous work has shown that the $HO_x$ and $NO_x$ measurements scale proportionally to the number of sparks in the packet (Jenkins et al., 2021), indicating that the chemicals generated by sparks within a packet are likely not overlapping. Second, due to the nonlinear chemistry between $HO_x$ and $NO_x$, we cannot assume that any modelling done with 10 sparks will scale simply to a single spark. Therefore, because each spark within a packet can be treated as an independent event, the modelling was done using $HO_x$ and $NO_x$ values scaled down to a single spark.


The initial OH and $HO_2$ determined from the extrapolation of the laboratory decays, scaled down 10-fold, were chosen as the initial OH and $HO_2$ (respectively) for the model runs. Using this same initial $HO_x$, three cases using different amounts of initial $NO_x$ were tested. In the first case, only the added NO was included in the model, and no spark $NO_x$ was included. In the second case, the added NO plus all the spark $NO_x$ was included, and in the third case, the added NO plus only a small percentage of

the spark $NO_x$ was included. The purified air used in the laboratory experiments was found to contain ~20 ppbv of CO (Thermo Scientific, 48i-TLE) which was also included in all the model experiments, along with wall loss at a rate of 0.9 $s^{-1}$ for OH (no wall loss was observed for $HO_2$). Model tests confirmed that even if up to 20 ppbv of $O_3$ (our limit of detection) had been generated in the laboratory experiments, it would not have significantly affected the $HO_x$ decays, so $O_3$ was not included in any of the model runs shown here. The model experiments were set to simulate 0.5 seconds of reaction time, enough to cover

the longest reaction timescale of the laboratory experiments, using the same pressure, temperature, and water vapor as the laboratory experiments, and included no dilution.

## 3 Results

### 3.1 Laboratory Results

As an increasing amount of NO was added to the air flow in the laboratory experiments, the OH and $HO_2$ decays became

progressively steeper, as shown Figure 3 (970 hPa and 360 hPa), Figure S2 (770 hPa and 570 hPa), and Figure S3 (average

slopes for all experiments). In other words, both OH and HO$_2$ decayed faster as more NO was added to the air flow. This dependence of the OH and HO$_2$ decays on the added NO indicates that LHO$_x$ is reacting mostly with the added NO, and little or not at all with the spark NO$_x$, supporting the hypothesis that the HO$_x$ we measure from spark and lightning discharges is produced separately from the spark NO$_x$. The average LNO$_x$ generated in the laboratory experiments is shown in Figure S4.

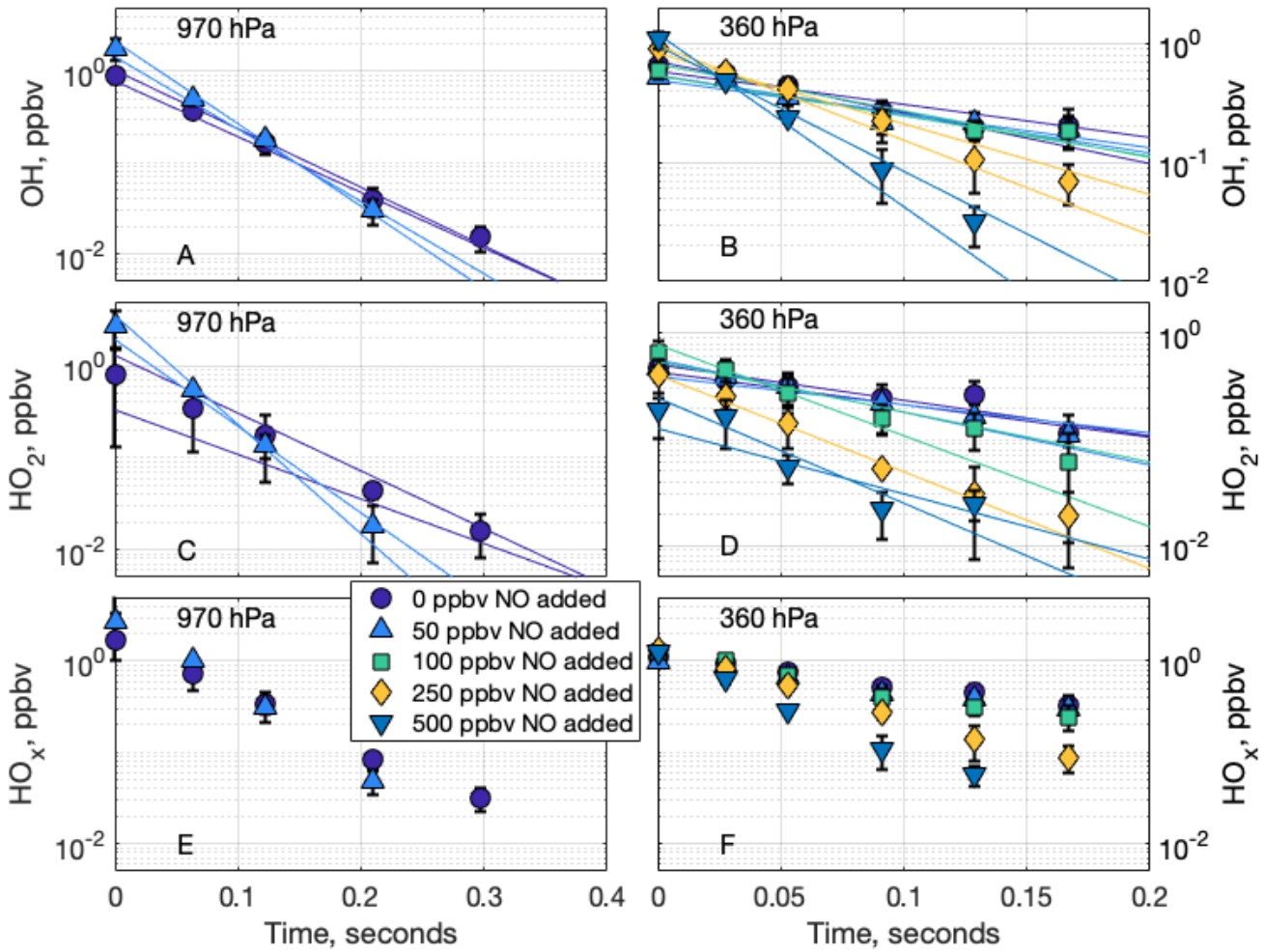


**Figure 3:** Laboratory decays of OH (A,B), HO$_2$ (C,D), and net HO$_x$ (E,F) at 970 hPa (A,C,E) and 360 hPa (B,D,F). The markers are the averaged data points containing 3 or 6 measurements from 1 or 2 laboratory experiments, respectively. The markers at time zero are the averaged extrapolated values from the decays. The lines on A, B, C, D are the linear fits to the individual decays. Error bars are the standard deviation from averaging the multiple laboratory measurements.


## 3.2 Laboratory versus model decays

Comparing the laboratory OH decays to the model decays from F0AM further supports the separate production of $LHO_x$ and LNO, but also indicates that $LHO_x$ and LNO or other chemical products from the spark discharges are likely interacting. For example, at 770 hPa and 0 ppbv of added NO, the laboratory $LHO_x$ measurements decay neither as fast as when 100% of the spark $NO_x$ is added to the model nor as slowly as when no spark $NO_x$ is added to the model (Figure 4A,B). If $LHO_x$ and $LNO_x$ were generated in the same place, the laboratory $LHO_x$ decays would match the model decay with 100% $LNO_x$ included, and if $LHO_x$ and $LNO_x$ did not interact at all, the laboratory decays would match the 0% $LNO_x$ model case. The laboratory decays falling in between the two model runs indicates that $LHO_x$ is either partially interacting with $LNO_x$, or it is interacting with some other product(s) from the sparks.

As the background NO was increased, the gap between the laboratory decay and 0% $LNO_x$ model case decreases (Figure 4C,D), and this gap decreases further as more background NO was added (Figure 4E,F). This decrease in the difference between the laboratory and model decays is likely because as the background NO was increased, it accounted for an increasing amount of the $HO_x$ reactivity compared to the spark products. This increasing agreement between the model and laboratory decays as the added NO increased can be seen at 970 hPa, 570 hPa, and 360 hPa as well (Figures S5, S6, S7, respectively), and is another indicator that $LHO_x$ is mostly made separate from the $LNO_x$ made in the spark hot channel.

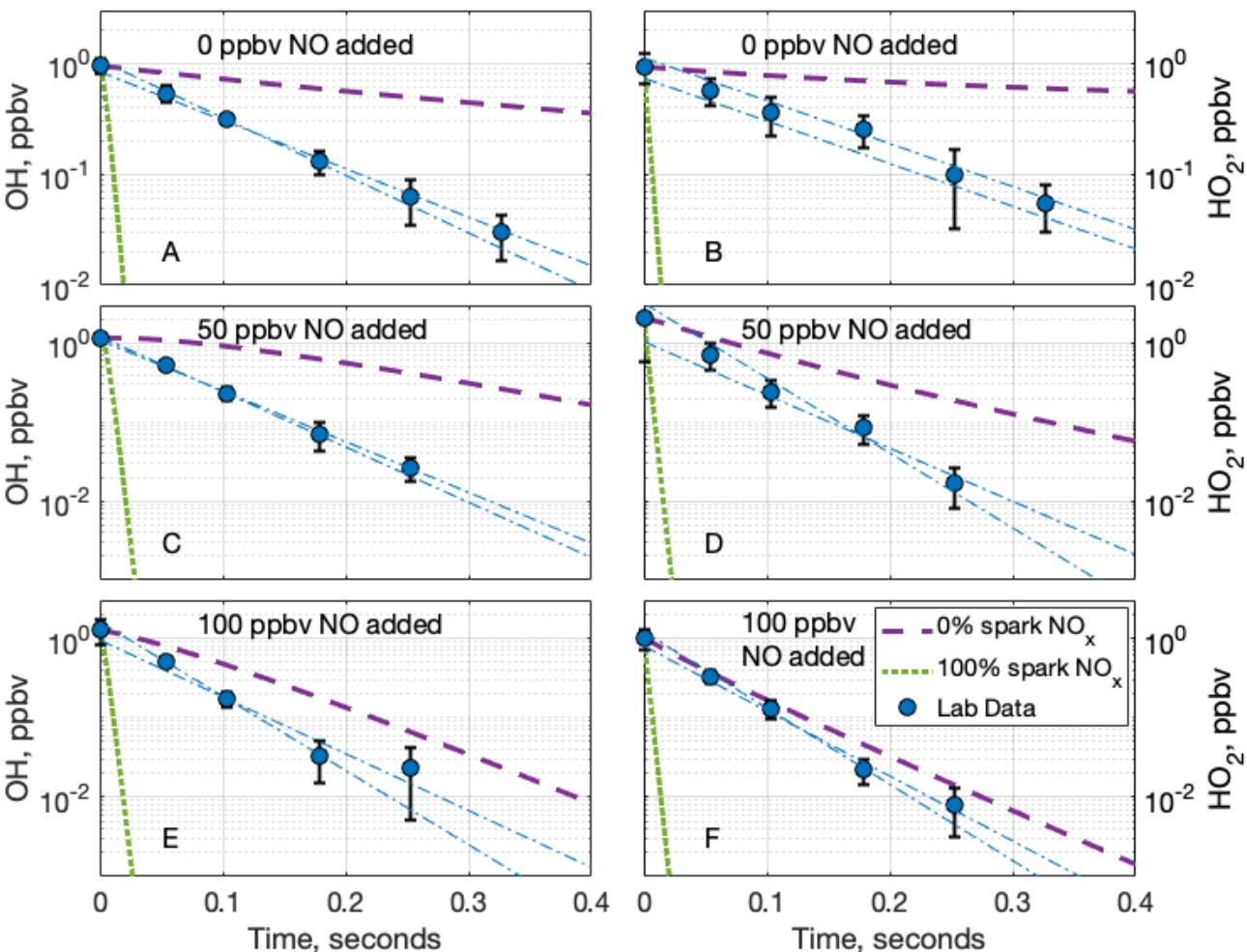

**Figure 4:** Comparison of measured OH (A,C,E) and HO₂ (B,D,F) laboratory decays and two model decays at 770 hPa and (A,B) 0 ppbv of added NO, (C,D) 50 ppbv of added NO, and (E,F) 100 ppbv of added NO. The dashed purple lines are the model decay with only the added NO, and includes no NO$_x$ from the spark, and the dotted green lines are the model decay with the added NO and all of the spark NO$_x$. The blue circles are the average laboratory measurements and average extrapolated value at time zero, while the dashed-dotted blue lines are the individual extrapolated linear fits to the laboratory data. Error bars are the standard deviation from averaging multiple measurements.

## 3.3 Improving the measurement-model agreement

The agreement between the laboratory and model decays is at its worst when 0 ppbv of NO was added in the laboratory experiments. As these cases are also the most relevant to the atmosphere, trying to resolve this disagreement can also give insight into lightning chemistry in the atmosphere.

When we first observed this measured-modelled discrepancy in Jenkins et al., 2021, we were able to resolve the discrepancy for both OH and $HO_2$ by including just 0.5% of the spark $NO_x$ in a model run. However, the model in the previous study was initialized using the full 10 spark packet data and also did not include the OH wall loss. Here, adding 3% of the spark $NO_x$ to the model (amounting to 61.6, 62.8, 69.9, and 90.7 ppbv of $NO_x$ at 970, 770, 570, and 360 hPa, respectively) brings agreement within uncertainty to the laboratory $HO_2$ data, but the OH data is still overestimated by the model (Figure S8). Adding 5% (104, 105, and 117 ppbv at 970, 770, and 570 hPa, respectively) or 10% (303 ppbv at 360 hPa) of $LNO_x$ instead brings measured-modelled agreement for OH, but the $HO_2$ data is then consistently underpredicted by the model (Figure S9). There is no amount of $LNO_x$ that can match the OH and $HO_2$ measurements simultaneously, leaving some chemistry still unaccounted for in the model.

Adding ~10 $s^{-1}$ of OH reactivity into the model along with 3% $LNO_x$ can resolve the discrepancy (Figure S10) within uncertainty. What chemical species could be responsible for this reactivity? In addition to the $HO_x$, $NO_x$ and $O_3$ we measure, many other species are generated in sparks as well, including atoms, ions, and excited states such as O, N, H, $N_2^+$, $O(^1D)$, $O^-$, and others; other molecules that are primary products of the discharge, like $N_2O$ and CO; and secondary products formed from reaction between or within the first two categories, like $H_2O_2$, HONO, and $NO_2$ (Bhetanabhotla et al., 1985; Boldi, 1992; Ripoll et al., 2014). For one (or more) of these species to account for the missing reactivity, it must fulfill a few criteria. First, its lifetime needs to be long enough so it is still present over the time frame we measure the $HO_x$ decays, at least 0.2-0.5 seconds post-discharge. Second, it needs to react with OH on the same 0.2-0.5 second time frame, so it must either react with OH quickly or be present in large enough quantities to compensate for a slow reaction rate. Third, it must spatially overlap with the $LHO_x$ we measure, so either it is produced in the corona sheath and/or UV radiation, or it is produced in large amounts in the hot core, with ~3% mixing out as we think $LNO_x$ is doing. Lastly, the reaction between OH and this species must not produce $HO_2$. The mismatch between the model and measurements is because OH is overpredicted by the model relative to $HO_2$. If the reaction between OH and the missing species yields $HO_2$, then instead of increasing the OH loss rate, OH will be quickly recycled through the reaction $HO_2 + NO \rightarrow OH + NO_2$.

Neither of the first two categories of species, the atoms, ions, and excited states or the other primary molecules, can account for the missing reactivity in the model. The lifetime of the atoms, ions, and excited states species will be too short to affect the $HO_x$ decays over 0.2-0.5 seconds, failing the first criterion. On the other hand, the primary products CO and $N_2O$ fail the second criterion. Both species are longer lived than the first category, but their reactions with OH are relatively slow, and not enough of these species will be produced to compensate. For example, only about ~340 ppbv of $N_2O$ is expected to be made in the combined hot core and corona sheath of a lightning flash (Brandvold et al., 1989; Brandvold et al., 1996; Donohoe et al., 1977; Hill et al., 1984; Levine et al., 1979), but ~11,000 ppmv would need to be produced in the laboratory sparks to compensate for a reaction rate of $k_{N_2O+OH} = 3.8 \times 10^{-17}$ $cm^3$ molecules$^{-1}$ $s^{-1}$ (Biermann et al., 1976). The reaction between CO

and OH is faster, with $k_{CO+OH}$ =2.3×10$^{-13}$ cm$^3$ molecules$^{-1}$ s$^{-1}$ at 970 hPa in F0AM, and only ~1.8 ppmv of CO is needed to satisfy the missing reactivity in the model. But this 1.8 ppmv is ~12% of the 14.6 ppmv of CO expected to be made in the lightning hot core (Bhetanabhotla et al., 1985; Levine et al., 1979), and it is unlikely that the laboratory sparks are making as

much CO as a lightning flash. The reaction of CO and OH also produces HO$_2$, leading to OH recycling.

The secondary discharge products are long-lived enough to still exist 0.2-0.5 seconds after the discharge, and their reaction rates with OH are faster than the rates with the primary products, so less of them are required to satisfy the missing reactivity compared to the primary products. Still, modelling results indicate that at most ~400 ppbv of H$_2$O$_2$ is generated in the lightning

hot channel, and if only 3% of the hot channel mixes out, then this will not be enough to satisfy the ~250 ppbv of H$_2$O$_2$ needed to account for the missing OH reactivity in the sparks based on the reaction rate of $k_{H_2O_2+OH}$ = 1.7×10$^{-12}$ cm$^3$ molecules$^{-1}$ s$^{-1}$ from F0AM. Additionally, the reaction of OH and H$_2$O$_2$ produces HO$_2$. For NO$_2$, we have already included 3% of what we measure in the laboratory experiments in the model runs, which amounts to <10 ppbv of NO$_2$.

HONO, however, could account for the missing reactivity. It meets all four of the criteria: it lasts long enough to affect the HO$_x$ decays; its reaction with OH does not recycle HO$_x$; it can react with OH over the 0.2-0.5 second time frame; and production of HONO in the core is expected to be high enough that only ~3% overlapping from the core could account for the OH reactivity. A model study including HONO production in the hot lightning core suggests as much as 12.6 ppmv of HONO can be generated within 10 ms of the discharge (Bhetanabhotla et al., 1985), and we only need ~70 ppbv of HONO to fulfill

the missing reactivity, using the F0AM reaction rate of $k_{OH+HONO}$=6.1×10$^{-12}$ cm$^3$ molecules$^{-1}$ s$^{-1}$. Even considering that the laboratory sparks are smaller and cooler than a real lightning flash, substantial HONO production in the range of 1-2 ppmv is possible for the laboratory sparks as well.

Chemical models of the hot lightning channel show that both LNO and LOH production is extreme inside the lightning hot

channel. For example, the model from Bhetanabhotla et al. (1985) has as much as 4300 ppmv of LNO and 860 ppmv of LOH initially produced, while the simulations of Ripoll et al. (2014) has as much as 42000 ppmv of LNO and 8400 ppmv LOH, with LNO and LOH within an order of magnitude of each other in the shock front. Little to no HO$_2$ is expected to be generated in the hot channel (Bhetanabhotla et al., 1985; Ripoll et al., 2014). As a test, a model experiment was run assuming 4 ppmv of LNO is initially produced in the laboratory sparks, which is only ~1.4-2 times our laboratory measurements for LNO, along

with 2.8 ppmv of hot core LOH and no other chemicals added. The result of this experiment is HONO production in the range of 1-2 ppbv across all pressures (Table 1). Additionally, this HONO is generated fast, before we make our first measurement of HO$_x$ in the laboratory flow tube. All the core LOH is also titrated to <1 pptv (our limit of detection in these experiments) over the same time frame the HONO is generated, so it would not be detected by GTHOS in the laboratory experiments. This model result is consistent with our laboratory observations because if substantial core LOH remains beyond the time the first

measurement is made in the laboratory, then we would expect to detect significantly more LOH than LHO$_2$ during the

experiments instead of the relatively equal amounts of LOH and LHO$_2$ that are actually detected. This result is also in line with the Bhetanabhotla et al. (1985) model prediction that all the core LOH should decay away very rapidly. The only model case where the core LOH is not titrated to less <1 pptv before the first laboratory measurement is made is at 360 hPa, but even at this pressure, the model predicts that HONO, NO, and NO$_2$ are all within 1% of their final values when that first measurement

is made.

Table 1. Comparison of the averaged NO and NO$_2$ measured in the laboratory experiments and the predicted NO, NO$_2$, and HONO from a model run starting with 4 ppmv of LNO and 2.8 ppmv of LOH.

| | 970 hPa | | 770 hPa | | 570 hPa | | 360 hPa | |
|---|---|---|---|---|---|---|---|---|
| | Lab | Model | Lab | Model | Lab | Model | Lab | Model |
| NO (ppbv) | 1850 | 1820 | 1950 | 1870 | 2200 | 1930 | 2900 | 2040 |
| NO$_2$ (ppbv) | 220 | 380 | 140 | 410 | 140 | 440 | 110 | 490 |
| HONO (ppbv) | - | 1670 | - | 1590 | - | 1490 | - | 1300 |
| Time* (s) | 0.064 | 0.019 | 0.055 | 0.027 | 0.042 | 0.0384 | 0.028 | 0.074 |

*For the laboratory data, time is when the first HO$_x$ measurement is made post-spark. For the model data, time is when OH
has been titrated to <1 pptv, our limit of detection in these experiments.

This model run demonstrates that HONO can be formed fast and in large amounts in the spark discharges. The initial chemistry in the sparks is occurring at thousands of degrees Celsius with electrons and many other chemical species besides NO and OH present, and the production of these species may have spatial dependencies that we cannot incorporate or account for in F0AM.
These limitations may explain why the model does not entirely reproduce the NO and NO$_2$ laboratory measurements. Still, the model results are within an order of magnitude of the laboratory results while simultaneously producing substantial HONO. Adding 3% of the modelled HONO from Table 1 into the model of the laboratory decays drastically improves the agreement between the modelled and measured OH, and in some cases brings the modelled and measured decays into agreement within the laboratory uncertainty (Figure 5). A diagram of the simplified HO$_x$ and NO$_x$ spark chemistry discussed in the preceding
paragraphs is shown in Figure 6.

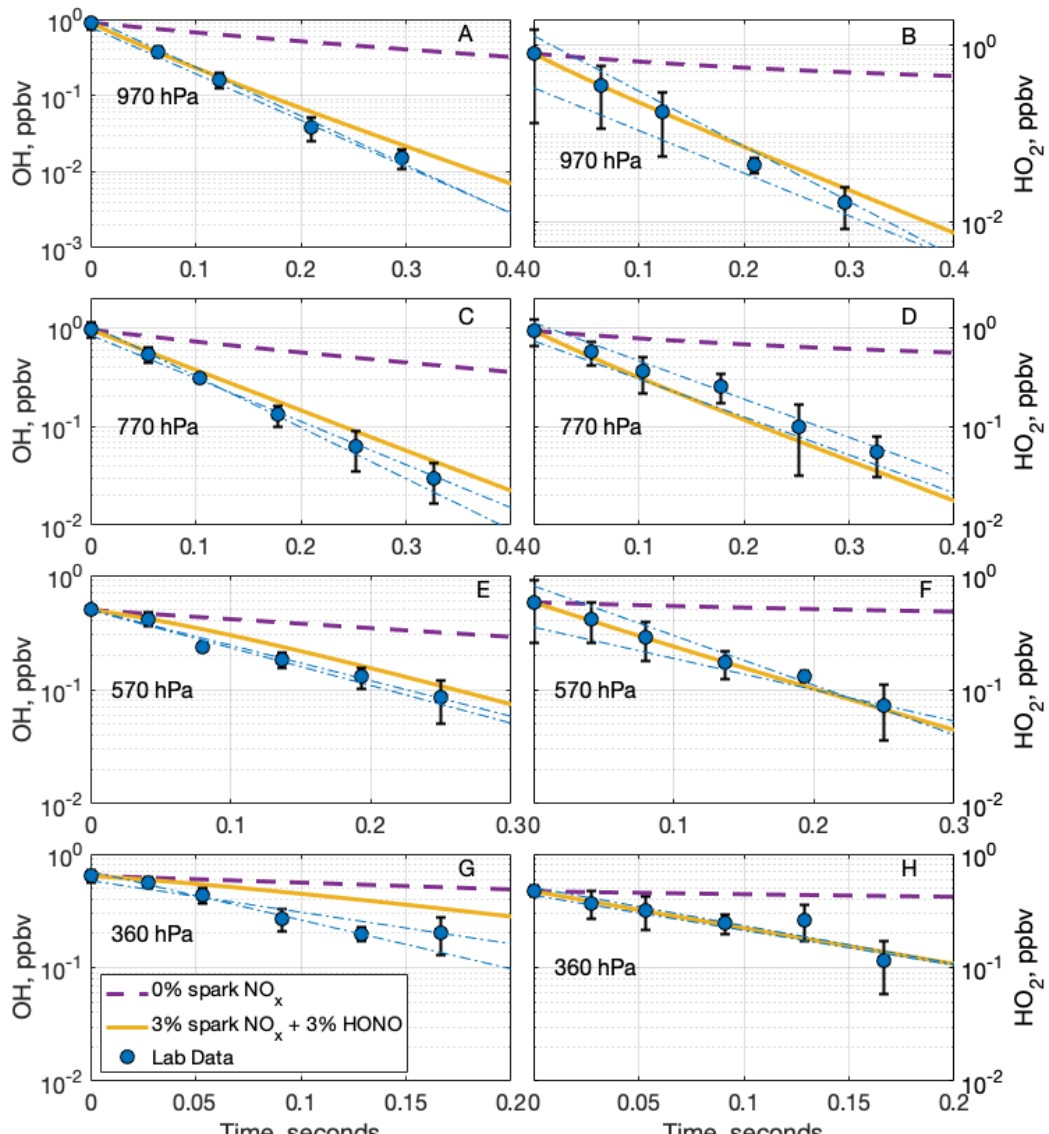

**Figure 5:** Comparison of measured OH (A,C,E,G) and HO$_2$ (B,D,F,H) laboratory decays and two model decays at (A,B) 970hP, (C,D) 770 hPa, (E,F) 570 hPa, and (G,H) 360 hPa. The dashed purple lines are the model decay including no NO$_x$ from the spark, and the solid yellow lines are the model decay including 3% the spark NO$_x$ and 3% of the HONO predicted to be generated in a model run. The blue circles are the average laboratory measurements and average extrapolated value at time zero, while the dashed-dotted blue lines are the individual extrapolated linear fits to the laboratory data. Error bars are the standard deviation from averaging multiple measurements.


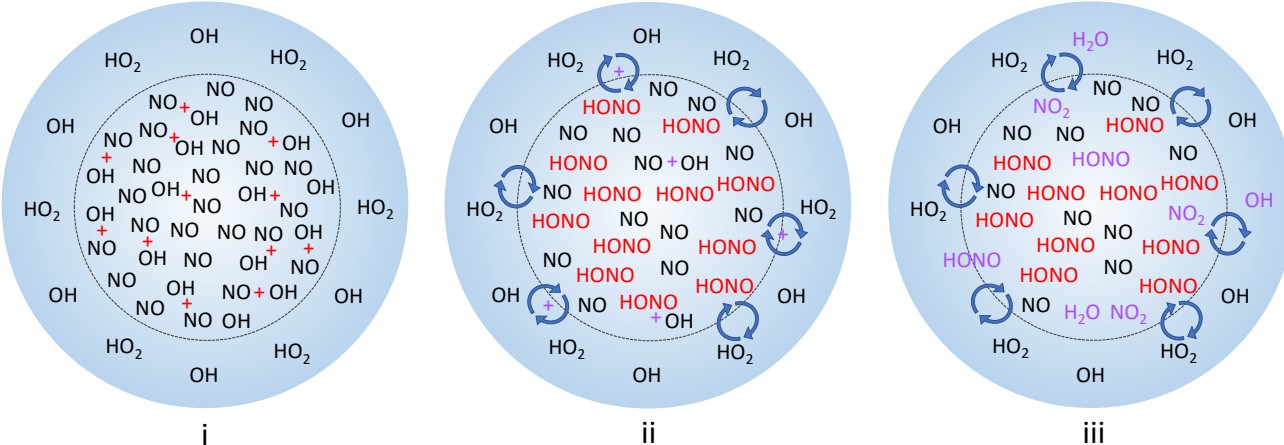

**Figure 6:** Simplified progression of the proposed $HO_x$ and $NO_x$ chemistry in spark and lightning discharges. (i) Initially, extreme amounts of NO and OH are made inside the lighting hot channel, indicated by the dashed inner circle, while OH and $HO_2$ are produced outside the hot channel in the corona sheath and UV radiation. (ii) The NO and OH in the hot channel react

and form HONO, while the species in the hot channel and corona sheath start to mix together. (iii) Inside the hot channel, any remaining OH reacts with NO and HONO, forming either more HONO or $H_2O$ and $NO_2$, respectively. Where the hot channel and corona sheath have started mixing, OH and $HO_2$ from the corona sheath react with NO from the core, forming HONO or OH and $NO_2$, respectively, while OH from the corona sheath and HONO from the core can also react to form $H_2O$ and $NO_2$.

## 4 Conclusions

Both the laboratory and model results across all the tested pressures confirm that the OH and $HO_2$ we measure from sparks are generated outside the lightning hot channel, separate from the core where the LNO is generated. It took 3% NO and 3% HONO to resolve the measured-modelled discrepancy in these laboratory experiments, where the sparks occurred in a flow tube with laminar flow and a fast air velocity. In the atmosphere, the percentage of NO or HONO reacting with $LHO_x$ could be lower or higher than 3%, depending on the turbulence and air velocity where the lightning flash occurs, and likely varies from one

lightning flash to the next. But the overall conclusion, that the $HO_x$ generated outside the hot channel only partially interacts with the hot channel products, will still be true in the atmosphere.

Additionally, these results indicate only that the substantial $LHO_x$ *we measure* is generated outside the hot channel; they do not imply that no $LHO_x$ is generated in the hot channel. As stated previously, modelling studies of the lightning hot channel

indicate that substantial $LHO_x$ is also generated in the hot channel, likely even more than we measure outside the hot channel.

But this hot channel HO$_x$ will be rapidly titrated away in the presence of the large NO also generated in the core, becoming substantial HONO. As for the LHO$_x$ we measure outside the hot channel, LHO$_x$ production has been found to be proportional to ultraviolet radiation (UV) production in corona discharge (Jenkins et al., 2022), and UV may also be responsible for the LHO$_x$ we measure in sparks and lightning. The consequence of this spatially separate production of LHO$_x$ and LNO is that LHO$_x$ is not immediately consumed by LNO in lightning flashes but instead is available to oxidize other pollutants in the atmosphere and contribute to global OH oxidation.


While we did not test the full range of possible tropospheric pressures and temperatures in this study, we still expect that these results apply for the lower pressures and lower temperatures found in the upper troposphere where most lightning occurs. Regardless of where it occurs in the troposphere, a lightning flash is composed of a hot core surrounded by a corona sheath and UV radiation, so HO$_x$ and NO$_x$ production is also expected to be spatially separate in the upper troposphere. Our previous study showed that the initial LNO$_x$ mixing ratio is independent of temperature and only slightly dependent on pressure, with less than a factor-of-2 difference in production between 970 hPa and 250 hPa, while the initial LHO$_x$ mixing ratio is independent of pressure and decreases with decreasing temperature, depending on the available water vapor (Jenkins and Brune, 2023). Therefore, we expect roughly the same LNO$_x$ production in the upper troposphere as was observed in the experiments here, with likely ~200-300 pptv of LHO$_x$ produced. The modelling results showed that for all the pressures tested in this study, the reaction $OH + NO + M \rightarrow HONO + M$ accounts for over half of the OH loss, while the reaction $HO_2 + NO \rightarrow OH + NO_2$ accounts for 80% of the HO$_2$ loss. The rates of these two reactions increases with decreasing temperature, although the rate of $OH + NO + M$ is also pressure dependent. However, further modelling tests using the lowered LHO$_x$ production with the same LNO$_x$ as was measured at 360 hPa demonstrate even at 200 hPa and 220 K, the reactions $OH + NO + M$ and $HO_2 + NO$ still accounts for more than 50% of the OH loss and 80% of the HO$_2$ loss, respectively. Thus, based on this information, we also expect the same subsequent HO$_x$-NO$_x$ chemistry to occur in the upper troposphere as shown for the pressures and temperatures here."




Differences in the model and laboratory HO$_x$ decays are resolved if substantial HONO is produced in the spark discharges and therefore HONO would also be a substantial product of lightning in the atmosphere. Aside from the substantial HONO production predicted in two lightning chemistry models (Bhetanabhotla et al., 1985; Hill & Rinker, 1981), enhanced HONO has been measured inside two different electrified convective clouds (Dix et a., 2009; Heue et al., 2014). The field studies estimate HONO mixing ratios from 37-160 pptv inside the clouds, a range much lower than what the modelling studies predict and the 1-2 ppbv we expect is made in the laboratory sparks. However, both field cases measured the HONO using differential optical absorption spectroscopy, generating long path measurements that are averaged over the entire length of the cloud, while higher resolution measurements would likely show very high HONO mixing ratios in lightning affected air and lower mixing ratios in air lightning did not pass through. We are not aware of any laboratory measurements of electrically produced HONO.


Measurements of electrically generated HONO, either in the laboratory or at higher spatial resolution in the field, would thus
be a good target for future work.

**Data Availability** All data shown in the figures is publicly available at Jenkins and Brune (2024).

**Author Contribution** Investigation, Methodology, Visualization, Original Manuscript Draft were by JMJ. Funding
Acquisition was by WHB. Conceptualization and Reviewing and Editing of the Manuscript were by WHB and JMJ.

**Competing Interests** The authors declare that they have no conflict of interest.

**Acknowledgements** We thank P. Stevens for lending us a microchannel plate detector after ours failed.

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
