# Peer review of "Spatially separate production of hydrogen oxides and nitric oxide in lightning"

_EGUsphere, 2024_

## Author Comment (AC1)

Reviewer comments and questions are shown in bold; responses are in plain text.

**The paper describes experiments to validate the hypothesis that NOx and HOx are generated spatially separated in lightning. This is an important aspect when trying to consider the impact of lightning induced HOx generation on the global oxidizing capacity. The experiments support the hypothesis, but I find that many details are missing or poorly described. Maybe it is necessary to read the earlier papers to understand, but I feel more information in the manuscript would make the paper much better.**

**Adding a schematic view of the experiment would be helpful (even if there is a figure in Jenkins 2021 it would be good to have one in this manuscript as well).**

A diagram of the experimental setup has been added as Figure S1 to the supplement. The subsequent supplemental figures have been renumbered in the text and supplement, and the following sentence has been added to text at Line 91: "A diagram of the laboratory setup is shown in Figure S1."

[Figure]

**Figure S1.** (A) Top-down diagram of the laboratory experimental setup showing the key components. (B) Side-view showing a close-up of the GTHOS inlet and Teflon line leading to the NOx and O₃ analyzers, which sample from the same volume as GTHOS in a 1.3-cm diameter tube place over the GTHOS inlet (shown as two horizontal lines), along with the relative

positions of the flow tube, copper rod, and tungsten electrode. Neither (A) nor (B) is shown to scale.

**And even after reading to the end of the paper, I am still wondering what you consider is the size of the core, ie. in your experiments what volume do you consider to contain NOx, and from which volume samples NOx and GTHOS?**

We do not have a way to directly measure the core size in our laboratory sparks, but the diameter is no bigger than ~1 mm, based on the diameter of the visible light generated. The sparks are only ~ 7 mm long, based on the gap between the electrodes. If we treat the spark as roughly cylindrical, the volume of the core is ~5.5 mm$^3$.

The GTHOS and the $NO_x$ and $O_3$ analyzers sample from the same volume, which is much larger than the spark size. Specifically, GTHOS has a sampling flow rate of ~4 standard liters per minute (slpm) or more depending on pressure, and the Teflon line leading to the $NO_x$ and $O_3$ analyzers, located ~2 mm downstream of the GTHOS opening, pulls ~13.5 slpm over to the analyzers, from which $NO_x$ and $O_3$ each sample 1 slpm. So all the analyzers sample from air that contains both the core and surrounding volume mixed together. There is also a short piece of Teflon tubing (1.3 cm diameter) affixed to the front of the GTHOS opening, which the $NO_x$-$O_3$ Teflon line points into, to ensure $HO_x$, $NO_x$, and $O_3$ are all sampled from the same volume of air.

When the spark occurs, the $NO_x$ is contained in the very small volume in the center and begins diffusing out. $HO_x$ is generated in the volume surrounding the center. This air containing $NO_x$ in the center and $HO_x$ in the surrounding volume travels together over to the analyzers. As the air travels forward, $NO_x$ diffuses out and reacts with the $HO_x$ generated in the surrounding air. For sparks generated in the same position, the $NO_x$ will diffuse out about the same amount and has the same amount of time to react with $HO_x$ before reaching the analyzers. As we move the spark back, $NO_x$ has more time to diffuse out and more time to react with $HO_x$, leading to the observed $HO_x$ decay. If the $NO_x$ and $HO_x$ were generated in the same place in the spark then we wouldn't detect the $HO_x$, because the amount of $NO_x$ generated in the spark would titrate all the $HO_x$ away before we could measure it. Thus, even though we are not sampling from only the core or only the surrounding air, we can still draw conclusions about where the $NO_x$ and $HO_x$ are being made in the spark.

The following has been added at Line 89: "A short piece of Teflon tubing (1.3 cm diameter x 2.5 cm long) was placed on the GTHOS inlet, and the opening of the Teflon tube leading to the $NO_x$ and $O_3$ analyzers was positioned ~2 mm downstream of the GTHOS opening and facing into the short piece of Teflon tubing. This arrangement

ensured that GTHOS and the $NO_x$ and $O_3$ analyzers all sampled from the same volume."

**For what reason did you do pressure dependent experiments? Initially I was thinking that you would look for increased diffusion of NOx out of the core volume with decreasing pressure, but you never draw any conclusions from the results obtained in pressure dependent experiments. Is the only difference the change in rate constants for pressure dependant reaction in MCM with decreasing pressure?**

We conducted experiments at different pressures because lightning can occur at any pressure in the troposphere, and we wanted to look for any pressure dependent effects in the chemistry. $NO_x$ will diffuse more as the pressure drops, and some of the reaction rates are pressure dependent, but ultimately, our conclusions were the same for all the pressures we tested in these experiments.

The sentences starting at Line 98 have been modified as follows (modifications are underlined): "Because lightning can occur at any pressure in the troposphere, data were collected at pressures of 970 hPa, 770 hPa, 570 hPa, and 360 hPa (all within ± 2%) to cover most of the tropospheric pressures. Data were also collected at water vapor mixing ratios between 2000-2400 ppmv and temperatures between 289-294K."

Line 287 has also been modified as follows (modification is underlined): "Both the laboratory and model results across all the tested pressures confirm…"

**Many details are missing: what is the time-gap between the different sparks, ie. what is the time window for the 10 sparks compared to the total measurement time such as shown in Figure 1?**

There is ~75 ms between each spark in the packet, so the entire 10 spark packet is completed within ~675 ms.

The following has been added at Line 81: "…with ~75 ms between each spark in the packet,…"

Figure 1 (and the other figures) is not showing total measurement time. It is showing how much time elapses between generating each spark and the measurement of $HO_x$ from each spark. For example, at 970 hPa, it takes ~64, 120, 210, and 290 ms for the $HO_x$ generated by the sparks to reach the GTHOS inlet from each of the positions, based on the distance from the discharge to the inlet and the 50 standard liter per minute air flow rate in the tube. So if a spark packet is generated at position 3 in the flow tube, the air exposed to the sparks will take 210 ms to reach GTHOS. It also means

the $HO_x$ generated by the spark has 210 ms of reaction time in the flow tube before we measure it.

**And at what repetition rate you generate spark packets?**

For each of the five positions in the flow tube, we generate four spark packets, with 5 second spacing between each packet. We then pause the sparking while the electrodes are pulled back in the flow tube to a new position. Once the electrodes have been repositioned, we again generate four spark packets with 5 second spacing. This process repeats for the remaining positions.

We have added the following to the manuscript at Line 94: "In each position, four spark packets were generated, with 5 second spacing between each packet."

**It would be interesting to see the raw data of the time evolution of OH and HO2 measurements for a spark packet.**

We have added Figure S2 to the supplement, which shows the OH and NO signals from the spark packets across each of the five positions.

We have also added to the main text at Line 108: "Figure S2 shows the OH and NO signals from the spark packets over time for one experiment."

Note that although we generate four spark packets in each position, there are signals from only 3 spark packets in the $HO_x$ data. For one of the spark packets, we switch the GTHOS laser to a different wavelength from the one OH absorbs at, to confirm that the OH and $HO_2$ signals do not contain any electrical interference from the spark.

The following has also been added at Line 94: "For one of the four spark packets, the laser on GTHOS was switched to a wavelength slightly off the OH absorption wavelength to confirm the absence of electrical interference in the OH and $HO_2$ signals."

[Figure]

**Figure S2.** Change in OH (A) and NO (B) mixing ratios due to the spark discharges at each of the five discharge positions at 770 hPa and 0 ppbv of added NO. Each peak is from one spark packet containing 10 sparks. OH and NO mixing ratios are indicated by the blue lines and use the y-axes on the left side of their respective subplots, while the distances from the discharge to the GTHOS inlet and Teflon line leading to the $NO_x$ analyzer are indicated by the orange lines and use the y-axes on the right side.

**How the GTHOS measurements have been synchronized with the discharge?**

There is no synchronization required. During the experiments GTHOS continuously makes measurements at a rate of 5 Hz, whether the discharge occurs or not. The $HO_x$ signals from the discharges just create spikes in an ongoing measurement.

**Over what time window do you integrate the peaks?**

For the OH and $HO_2$ measurements, we integrate the signal from each 10-spark packet over 2.2 seconds. The actual OH and $HO_2$ peaks are only ~1.2 second wide, but we use

a broader window in the data processing to ensure we capture the whole peak. The $HO_x$ measurements are made rapidly, so the $HO_x$ signals from the spark packet do not spread out much.

For NO and total $NO_x$, we integrate over 4.8 seconds, which is about the whole width of these peaks. The air travels through a Teflon line and a filter before reaching the $NO_x$ analyzer, which spreads out the NO and $NO_x$ samples along the way. As a result, these signals come out broader than the $HO_x$ signals, and we integrate over a bigger window.

The following has been added starting at Line 109: "For the OH and $HO_2$ measurements, the peaks were about ~1 second wide and were integrated over 2.2 seconds, while the NO and $NO_x$ peaks were ~4.8 seconds wide and were also integrated over 4.8 seconds."

**Over what distance do you move the discharge?**

We move the discharge over a total distance of 27.5 cm, starting at 5.5 cm from the GTHOS inlet.

The following has been added to the manuscript at Line 94: "...over a total distance of 27.5 cm."

**What is the time resolution of the NOx analyser and how well resolved are the NOx measurements for one spark packet?**

The $NO_x$ analyzer collects data at a rate of 2 Hz.

The following sentence has been added at Line 82: "The $NO_x$ analyzer collected data at a rate of 2 Hz and the $O_3$ analyzer collected data at a rate of 1 Hz."

We cannot distinguish the signals from each individual spark in the 10-spark packet. If we could reliably detect the signal from a single spark on GTHOS and the $NO_x$ and $O_3$ analyzers, we would just use single sparks instead of the spark packets. The reason for using 10 sparks has to do with generating a $HO_x$ signal that is broad and intense enough for GTHOS to pick up consistently.

Essentially, the actual signal from a 10-spark packet for both $HO_x$ and $NO_x$ is a peak from the spark, following by a dip from the 75 ms gap between sparks, followed by another peak from the next spark, and so on. The air between sparks travels at least ~6.5 cm, which is enough distance so the that chemicals from one spark either do not overlap at all or overlap negligibly with the next. But because the instruments cannot

resolve the signals to this extent, we instead detect $HO_x$ and $NO_x$ signals that are a single peak, the accumulation of the entire 10 spark packet.

**You say: " so any change in the NOx mixing ratio across the different positions was assumed to come from diffusion and not chemical loss. ": can you give an order of magnitude of the NOx loss over the different positions, maybe show a figure? At different pressures?**

The change in $NO_x$ mixing ratio decreases from the first position to the last position a factor of around 1.8 to 3.4, depending on pressure. This decrease is consistent with diffusion of the core over time.

A new Figure S3 has been added to the supplement showing the average change in $NO_x$ over the different discharge positions, for all four pressures tested in the laboratory experiments.

The following has also been added to the text starting at Line 114: "The average change in $NO_x$ over the different discharge positions in the flow tube is shown in Figure S3 for all four pressures tested."

[Figure]

**Figure S3.** The average $NO_x$ measured for the full 10-spark packet at each position in the flow tube for the experiments with no added NO. Error bars are the standard deviation from averaging the multiple measurements. Distance to the inlet is the distance from the opening of the Teflon tube that leads to the $NO_x$ analyzer to the discharge in the flow tube.

**How do you calibrate your HOx measurements?**

We pass air at a known flow rate and with a known concentration of water vapor through 185 nm radiation from a mercury lamp. The 185 nm radiation photodissociates the water vapor and generates equal amounts of OH and $HO_2$ in the air, which is sent directly into the GTHOS inlet. Along with the water vapor concentration, we know the flux from the lamp, the exposure time of the air to the lamp (because we know the flow rate), the quantum yield, and the photolysis cross section of water vapor, so we know exactly how much OH and $HO_2$ is formed in the calibration system and thus measured by GTHOS. We vary the water vapor concentration to vary the OH and $HO_2$ signals during the calibration. We also use inlets with different sized openings to create different internal pressures in the GTHOS system and mimic the different pressures GTHOS encounters in the laboratory

experiments. For further details on GTHOS and its calibration procedures, please see Faloona et al., 2004, which is referenced in the manuscript. Because GTHOS has been in regular use for over 20 years now, and the details of its functioning and calibration are well-documented in Faloona et al. and many other publications, we will leave out discussion of the GTHOS calibration procedure in this manuscript. This calibration method is not unique to GTHOS – essentially every other group that measures OH uses some version of it.

**What is the HO2 conversion factor within GTHOS?**

Because these experiments were conducted without hydrocarbons and therefore there is no possibility of an $RO_2$ interference in the $HO_2$ measurement, we set the NO flow to maximize the conversion of $HO_2$ to OH, which based on the $HO_x$ signals during calibration and is greater than ~95%.

**Line 142: what do you means with "The model experiments ran for 0.5 seconds of experiment time"? How long was the experiment time?**

Because all our laboratory measurements were made with a total reaction time of less than 0.5 seconds, we had the model experiments in F0AM simulate 0.5 seconds of chemistry. This time is not the model computation time.

This part of Line 142 in the text has been reworded to: "The model experiments were set to simulate 0.5 seconds of reaction time, enough to cover the longest reaction timescale of the laboratory experiments,…"

**I understand that the conclusion is, that NOx is generated spatially different from HOx. My question is how do you take this into account when you measure HOx and NOx? Because it should be very sensitive to the position of the sampling points for GTHOS and the NOx analyser? Maybe add a Figure 4 type where you indicate the (estimated) size of the core as well as the precise position of the NOx and HOx sampling. Would it be possible to move the sampling points? Maybe at least for the NOx analyser?**

In this experiment, it is not possible to sample the core and the surrounding air individually because, as detailed in a previous answer, the core is much smaller than the sampling volume of the instruments, and really any such air sampling instrument used by atmospheric chemists.

To add to our previous discussion of the sampling, we do align the spark in the center of the flow tube horizontally and vertically, which is also in line with the center of the GTHOS inlet and opening of the Teflon line leading to the $NO_x$ and $O_3$ analyzers to within ~1 mm. Please see Figure S1b for a diagram showing the relative positions of the inlet, Teflon line, and spark. But the large sampling volume relative to the spark volume means even if it the spark gets a little off center, our results are unaffected. Thus, we are not very sensitive to the sampling positions for the analyzers. Even if we did move the sampling points (which is not possible for GTHOS), it would not affect any of our conclusions.

**You correct both species for 15% due to not perfect sampling and diffusion, but I do not clearly understand how you can deduce from the NOx measurement that the correction factor should be the same for the HOx measurements.**

We do not deduce from the $NO_x$ measurements that the correction factor should be the same for the $HO_x$ measurements. We deduce the correction factors are about the same between $NO_x$ and $HO_x$ because "OH and $HO_2$ have similar diffusion coefficients to $NO_x$" (Line 111). Because $HO_x$ and $NO_x$ have similar diffusion coefficients, we expect them to diffuse at a similar rate in the flow tube, and therefore we can apply the same diffusion corrections to both the $HO_x$ and $NO_x$ data. We cannot measure the $HO_x$ sampling loss directly like we can for $NO_x$, so we make a reasonable assumption about what is happening with $HO_x$.

A reference has been added to the sentence starting at Line 111: "OH and $HO_2$ have similar diffusion coefficients to $NO_x$ (Tang et al., 2014), so OH and $HO_2$ were also corrected up 15% to account for sampling." This reference has also been added to the reference list: Tang, M. J., Cox, R. A., and Kalberer, M.: Compliation and evaluation of gas phase diffusion coefficients of reactive trace gases in the atmosphere: volume 1. Inorganic compounds, Atmos. Chem. Phys., 14, 9233-9247, doi:10.5194/acp-14-9233-2014, 2014.

**When you say " The laboratory air was found to contain ~20 ppbv of CO " I guess you mean the purified air that entered the reactor?**

Correct.

This Line has been modified to: "The purified air used in the laboratory experiments was found to contain ~20 ppbv of CO (Thermo Scientific, 48i-TLE)…"

**How did you measure and can you safely assume the absence of other trace gases in the purified air? As any remaining VOCs could generate many different**

**species in the discharge, that could react with OH, it seems to me important to verify the quality of the purified air.**

We can safely assume the presence of any trace gases is minimal and negligible to our experiments for the following reasons:

1. In our first paper using this laboratory setup, we measured the OH reactivity of the purified air, and found it be ~1 s$^{-1}$. This reactivity is equivalent to the reactivity in the relatively clean free troposphere, where most lightning occurs (Jenkins et al., 2021). Since this initial study, there have been some upgrades to our air compressor and dryer that have further decreased this reactivity.

2. This decrease in reactivity was confirmed in a recently published paper where we used the same laboratory setup and purified air described in this paper to conduct a kinetic experiment looking at the OH + HO$_2$ reaction rate. In this kinetics paper, we found the OH reactivity due to impurities in our purified air to be ~0.35 s$^{-1}$ (Brune & Jenkins, 2024). So we know the air we use the experiments is quite clean.

3. Lastly, even if we had a rather large impurity, its impact on the OH decays on the <0.5 second time scales we observe them over in the laboratory would be minimal. For example, if we run a version of the 3% NOx + 3% HONO model run that also includes 20 ppbv of formaldehyde — a large amount of formaldehyde that would only be present in a very polluted city and is certainly not present in our laboratory — the OH decays would not change much, as shown below.

[Figure]

For these reasons, we are confident that the purified air is very clean and there are no impurities interfering without results.

The following has been added at Line 75: "Purified and dried air, with an OH reactivity of ~0.35 s$^{-1}$ (Brune & Jenkins, 2024), was flowed through a bubbler…"

The Brune & Jenkins 2024 reference has also been added to the reference list: Brune, W. H., and Jenkins, J. M.: Is the reaction rate coefficient for $OH + HO_2 \rightarrow H_2O + O_2$ dependent on water vapor?, JACS Au, 4, 4921–4926, doi: 10.1021/jacsau.4c00905, 2024.

**Figure 1: why are there no measurements with higher NOx at the higher pressure? Any good reason? Because you say "OH and HO2 decays became progressively steeper, as shown Figure 1 (970 hPa and 360 hPa)" and to be really convincing it would have been interesting to see the trace at 970 hPa for 100 ppb NO at least. Did you not do the measurements at higher NO or were there not**

**enough data points available at higher NO (even though the 50ppb data look good enough to still expect good data also at 100ppb)?**

The reason is as we explain at Line 118: "In some experiments, the $HO_x$ decay was fast enough that usable $HO_x$ data was not available at all 5 flow tube positions. If at least 3 positions had clear OH and $HO_2$ signals, the decay was included in the results; if only 2 positions or less were available, the data were not used in the results, as there was not enough confidence in the extrapolated fit." We did conduct experiments with added NO amounts of 100 ppbv and 250 ppbv at 970 hPa, but we only had good data at two positions and one position, respectively, because the decays were so rapid. So yes, we made the measurements, and we did not include them for the reason we stated at Line 118.

If you set aside the extrapolated data point at time zero, there were good data at four positions in the 0 ppbv case and good data at three positions in the 50 ppbv data at 970 hPa. So it is not surprising that we only had good data at two (one) positions at 100 ppbv (250 ppv) of added NO, which was not enough to fit a curve.

We have modified Line 118 as follows (modifications are underlined): "In some experiments, the $HO_x$ decay was fast enough that the $HO_x$ data became too small and imprecise to use at farther discharge positions in the flow tube."

**Even if it seems clear, maybe explain in the legend that HOx is the sum of OH and HOx, ie. the total signal together with (in the main text) some information on the calibration procedure of the HOx signal.**

We define $HO_x$ as the sum of OH and $HO_2$ in the Introduction, at Lines 28-29 "OH and $HO_2$ (hydrogen oxides or $HO_x$)"; we now modify this sentence to the following: "OH and $HO_2$ (together called the hydrogen oxides or $HO_x$)" for more clarity. Because we do define the $HO_x$ abbreviation in the Introduction, we will not define it again in the legend. Additionally, for the reasons stated previously, we will not discuss the GTHOS calibration in this manuscript.

**Line 161 and 168: I guess you talk about the Figure 2 and not Figure 3?**

Yes, these Lines have been corrected to refer to Figure 2 instead of Figure 3.

**It would be good to give the absolute concentration of spark NO and not just giving the % so the reader can have an idea of how much this is compared to the added NO.**

Lines 188-191 have been modified as follows to include $NO_x$ mixing ratios (modifications are underlined): "Here, adding 3% of the spark $NO_x$ to the model (amounting to 61.6, 62.8, 69.9, and 90.7 ppbv of $NO_x$ at 970, 770, 570, and 360 hPa, respectively) brings agreement within uncertainty to the laboratory $HO_2$ data, but the OH data are still overestimated by the model (Figure S10). Adding 5% (104, 105, and 117 ppbv at 970, 770, and 570 hPa, respectively) or 10% (303 ppbv at 360 hPa) of $LNO_x$ instead brings measured-modelled agreement for OH, but the $HO_2$ data are then consistently underpredicted by the model (Figure S11)."

**Line 248: The sentence "All the core LOH is also titrated to <1 pptv (our limit of detection in these experiments) over the same time frame the HONO is generated, so it would not be detected by GTHOS in the laboratory experiments, consistent with our observations." is not clear to me: I understand that you cannot make a difference between core and outer LOH in the measurement, so what is consistent with your observations?**

It is true that if both core and outer LOH are present in the sample, it would all be mixed together and we would be unable to distinguish how much comes from inside vs. outside the core. But there should not be core LOH present in the sample at all; it should all decay away before we can measure it, leaving only the outer LOH.

In the laboratory experiments, we measure relatively equal amounts of LOH and $LHO_2$, and we make our first measurement 28-64 ms after the discharge, depending on pressure.

Modelling studies of the lightning hot channel or core have shown that:

1. Extremely large amounts of LOH is made in the lightning core, but the $LHO_2$ made is orders of magnitude less (Bhetanabhotla et al., 1985; Ripoll et al., 2014).
2. This core $LHO_x$ should decay away within 10 ms (Bhetanabhotla et al., 1985).

Based on these modelling studies, we would not expect any core $LHO_x$ to remain by the time we are able to make our first measurements. Therefore, when we do our model run in F0AM to check the feasibility of forming enough HONO, any core LOH we add to this model run must all decay away in less than 28-64 ms, or before we are able to make an $LHO_x$ measurement in the laboratory. Should the core LOH remain after this timeframe, we would be able to detect it because LOH would be significantly greater than $LHO_2$, inconsistent with our observations and the lightning model predictions from Ripoll et al. and Bhetanabhotla et al. So the fact that all the core LOH we use in the model run is gone by the time we would start making measurements in the laboratory is consistent with our observations.

This Line has been modified and new sentences have been added as follows: "All the core LOH is also titrated to <1 pptv (our limit of detection in these experiments) over the same time frame the HONO is generated, so it would not be detected by GTHOS in the laboratory experiments. This model result is consistent with our laboratory observations because if substantial core LOH remains beyond the time the first measurement is made in the laboratory, then we would expect to detect significantly more LOH than $LHO_2$ during the experiments instead of the relatively equal amounts of LOH and $LHO_2$ that are actually detected. This result is also in line with the Bhetanabhotla et al. (1985) model prediction that all the core LOH should decay away very rapidly."

The following has also been added at Line 243: "Little to no $HO_2$ is expected to be generated in the hot channel (Bhetanabhotla et al., 1985; Ripoll et al., 2014)."

**Line 81 : Should probably be 5 kHz.**

5 Hz is correct; GTHOS collects data at a rate of 5 times per second.

**The scaling in Figure 3c is missing.**

Figure 3 has been revised to include the scaling on subplot C.

Figure S4 has also been revised to correct the marker shapes on subplot A.

---

## Author Comment (AC2)

Reviewer comments and questions are shown in bold; responses are in plain text.

**In "Spatially separate production of hydrogen oxides and nitric oxide in lightning", Jenkins et al. report the generation of OH from lightning (LOH) based on laboratory studies and F0AM box model simulations. The authors suggest that OH can persist in the atmosphere as it is generated in the corona sheath, which is spatially separated from the lightning core, where large amounts of NO titrate OH immediately. They further propose that large amounts of HONO are generated in lightning strokes from the reaction of NO with OH.**

**This research is highly important given the limited knowledge of lightning and particularly the role of OH and suits the scope of ACP. However, I have several important questions which need to be addressed before I can recommend this manuscript for publication. While the introduction is well written and can be followed easily, the methods and results parts need some restructuring and additional information, as it is sometimes difficult to follow the reasoning. The authors often refer to their previous studies - I recommend adding a paragraph on these results as they are important for this manuscript, but not all readers might be familiar with them. Further, a schematic of the experimental set up could help to understand the laboratory methods better. I additionally wonder if the authors could carry out experiments under upper tropospheric conditions, where temperature, pressure and water vapor concentrations are different than those pursued by the authors, but lightning is most relevant. How relevant are the experiments to the actual conditions of lightning in the atmosphere (mostly in the UT)? Please find my specific comments and questions in the following.**

**Specific comments:**

**Line 33 & 46: 100s and 10s could be mistaken for 100 and 10 seconds. Maybe this could be spelled out.**

'100s' has been changed to 'hundreds' and '10s' has been changed to 'tens' in the manuscript.

**Lines 53 ff: "UV radiation can also make extreme OH…" As far as I understood the authors earlier, the UV radiation is created from the corona. However, here it sounds like corona and UV radiation represent to different mechanism for creating LOH / LHOx. Could you please clarify this?**

In the previous paragraph, we do say that both the corona and UV radiation extend around the hot core, but we do not say anywhere that the corona creates the UV

radiation, so we are unsure what we stated that caused this misunderstanding. Both the hot core and corona generate UV radiation, which can extend out beyond either the hot core or corona sheath.

We have added the following sentence at Line 45: "This UV radiation is generated by both the hot core and the corona sheath."

**And what would be the mechanism of HOx formation from corona?**

Aside from the UV radiation, there are several different reactions that generate $HO_x$ in corona discharges. The table in Bruggeman and Schram (2010) (referenced in the manuscript) shows these pathways, most of which are initiated by an electron impacting a molecule or atom. Some examples include:

$electron + H_2O \rightarrow OH + H$ followed by $H + O_2 + M \rightarrow HO_2 + M$

$O^1D + H_2O \rightarrow 2OH$

$H_2O^+ + electron \rightarrow OH + H$ followed by $H + O_2 + M \rightarrow HO_2 + M$

$H + O_2 \rightarrow OH + O$

The following sentence has been added at Line 53: "These pathways include an electron dissociating a water molecule ($electron + H_2O \rightarrow OH + H$) or an excited state oxygen atom dissociating a water molecule ($O^1D + H_2O \rightarrow 2OH$), among several others."

**How much HOx is formed in the corona versus from UV radiation?**

We investigated how $HO_x$ is formed in corona discharges in Jenkins et al., 2022. This previous study only looked at corona discharges, not sparks. Non-UV pathways for making $HO_x$ in corona discharges are initiated by electrons from the corona discharge colliding with molecules or atoms. The number and energy of these electrons is different in positive and negative corona, so if these non-UV pathways were dominate, our $HO_x$ production should have had a polarity dependence (much like $O_3$ production in corona discharges does). But our observations showed no difference in $HO_x$ production between positive and negative polarity corona discharge. Based on this information, along with the relatively equal amounts of OH and $HO_2$ produced, we concluded that most of the $HO_x$ was being made by the 185 nm UV radiation emitted by the corona.

We are currently collecting UV data from spark discharges, so we are in the process of performing a similar analysis for spark discharges. At this point, we cannot rule out that a reaction sequence such as $electron + H_2O \rightarrow OH + H$ followed by $H + O_2 + M \rightarrow HO_2 + M$ could be a more important contributor to $HO_x$ production in sparks/lightning compared to pure corona discharge, as sparks/lightning are not a single polarity but alternate between positive and negative. So we do not know how important UV versus non-UV pathways are for $HO_x$ production in sparks (yet).

**Line 61: "spatially separation LHOx and LNO production is possible" Is there a word missing? Or "spatially separate"?**

No words are missing; it is just a typo. "spatially separation" has been corrected to "spatially separate".

**Line 64: A previous reviewer of this manuscript or of another paper? If it refers to a different study, I recommend removing "as suggested by a previous reviewer" and replacing it by a reference to the study it follows up on.**

We were referring to a reviewer of our first paper on the subject of $LHO_x$, Jenkins et al., 2021. The phrase has now been removed.

**Lines 73 ff.: It could be helpful for the understanding to add a schematic of the experimental set up, including the location of the sampling and the positions of the discharge generation.**

A diagram of the laboratory setup has been added as Figure S1 to the supplement, which shows the position of the inlet relative to the sparks in subdiagram (B). The following sentence has been added to text at Line 91: "A diagram of the laboratory setup is shown in Figure S1."

The distance between each discharge position and inlet are shown in the new Figures S2 and S3 that were added in response to Reviewer 1's questions, so are not included in the diagram in Figure S1. The distances are 5.5, 10.5, 18, 25.5, and 33 cm.

[Figure]

**Figure S1.** (A) Top-down diagram of the laboratory experimental setup showing the key components. (B) Side-view showing a close-up of the GTHOS inlet and Teflon line leading to the $NO_x$ and $O_3$ analyzers, which sample from the same volume as GTHOS in a 1-cm diameter tube place over the GTHOS inlet (shown as two horizontal lines), along with the relative positions of the flow tube, copper rod, and tungsten electrode. Neither (A) nor (B) is shown to scale.

**Line 76: Is the flow tube "wide enough" to capture the center of the spark and the corona individually?**

Doing this kind of sampling is not possible for us presently, regardless of the flow tube width. The core and corona start mixing very quickly, so we would need to sample the air right where the spark occurs if we wanted to sample the center and corona individually. However, we cannot sample closer than ~5.5 cm from the sparks because otherwise the sparks could strike the instrument inlets and cause electrical interference in the signals or damage the instruments.

The other issue is that the instruments need rather large flows relative to the size of the sparks. The sparks are 0.7 cm long, but the GTHOS sampling flow rate alone is ~4 standard liter per minute (slpm) or greater, and the $NO_x$ and $O_3$ sampling flow rates are each ~1 slpm. So the size of the sparks would also need to be scaled up to only sample

the center or corona (which would also likely create more electrical interference issues).

**Line 96: What's the mixing ratio of NO in the flow tube generated from the spark?**

The generated $LNO_x$ varied from ~2-3 ppmv per spark. A new figure and caption, shown below, has been added to the supplement as Figure S6 and shows the average $LNO_x$ generated in each of the laboratory experiments.

The following line has also been added to the main text: "The average $LNO_x$ generated in the laboratory experiments is shown in Figure S6."

[Figure]

**Figure S6.** Average $LNO_x$ generated per spark in the laboratory experiments at different pressures and added NO amounts. Error bars are the standard deviations from multiple (19-20) measurements.

**Line 99: Have you tried to carry out an experiment under upper tropospheric conditions, i.e. lower temperature (e.g. 220K) and lower pressures (e.g. 200hPa),**

**where lightning is most frequent? If that's not possible with your set-up I recommend discussing the implications of different conditions. Reaction (R2) seems to be highly dependent on temperature and pressure**

They were not NO addition experiments like in this study, but yes, we did experiments at temperatures as low as ~260K and pressures down to ~250 hPa in a previous study (Jenkins and Brune, 2023). These values are the lowest we can do with the current laboratory setup. We found that decreasing temperature can decrease the $LHO_x$ production (the exact decrease is dependent on water vapor), but lower pressure does not significantly affect the $LHO_x$ production. $LNO_x$ production increases slightly with lower pressure (less than a factor of 2 from 970 hPa to 250 hPa) and is independent of temperature.

For this study, we are more interested in the $HO_x$ decays rather than the absolute $HO_x$ or $NO_x$ production. The reactions $OH + NO + M \rightarrow HONO + M$ and $HO_2 + NO \rightarrow OH + NO_2$ are by far the most dominant reactions at all the pressures we tested, accounting for 50% or more of the OH losses and 80% or more of the $HO_2$ losses, respectively. For the 3% NO + 3% HONO runs, $OH + HONO \rightarrow H_2O + N_2O$ was the second largest OH loss and $OH + HO_2 \rightarrow H_2O + O_2$ was the second largest $HO_2$ loss. The rate of all these reactions increases ~30% as the temperature decreases from 290 K to 220 K, except for $OH + NO + M$, which increases ~110%. $OH + NO + M$ is also the only reaction with a rate coefficient dependent on pressure, so some of the rate increase from the temperature drop will be offset by the decrease in pressure in the upper troposphere. However, modelling tests show that $OH + NO + M$ would still be expected to be the dominant OH loss reaction at 200 hPa and 220 K, followed by $OH + HONO$. Similarly, $HO_2 + NO$ remains as the dominant $HO_2$ loss followed by $OH + HO_2$. Based on this information, we would not expect any significant changes to the $HO_x$-$NO_x$ chemistry at lower pressures and temperatures.

Reaction R2 does have temperature and pressure dependences, but it is still a fast reaction for any temperature/pressure found in the troposphere. The latest JPL* evaluation lists the rate coefficient for R2 at 298 K and 1 atm (near surface conditions) as $1.15 \times 10^{-12}$ $cm^{-3}$ molecule$^{-1}$ s$^{-1}$. Using the low-pressure limit equation in the JPL with a pressure of 200 hPa and temperature of 220K, the rate coefficient becomes $6.0 \times 10^{-13}$ $cm^{-3}$ molecule$^{-1}$ s$^{-1}$, a decrease of about ~50%. If you add in the uncertainties to these values (at least ±20% at 298 K), then that 50% difference becomes even less significant. Thus, despite appearances, R2 is really not very dependent on temperature or pressure for the ranges found in the troposphere.

We have added the following paragraph to the Conclusion in the main text, at Line 298: "While we did not test the full range of possible tropospheric pressures and

temperatures in this study, we still expect that these results apply for the lower pressures and lower temperatures found in the upper troposphere where most lightning occurs. Regardless of where it occurs in the troposphere, a lightning flash is composed of a hot core surrounded by a corona sheath and UV radiation, so $HO_x$ and $NO_x$ production is also expected to be spatially separate in the upper troposphere. Our previous study showed that the initial $LNO_x$ mixing ratio is independent of temperature and only slightly dependent on pressure, with less than a factor-of-2 difference in production between 970 hPa and 250 hPa, while the initial $LHO_x$ mixing ratio is independent of pressure and decreases with decreasing temperature, depending on the available water vapor (Jenkins and Brune, 2023). Therefore, we expect roughly the same $LNO_x$ production in the upper troposphere as was observed in the experiments here, with likely ~200-300 pptv of $LHO_x$ produced. The modelling results showed that for all the pressures tested in this study, the reaction $OH + NO + M \rightarrow HONO + M$ accounts for over half of the OH loss, while the reaction $HO_2 + NO \rightarrow OH + NO_2$ accounts for 80% of the $HO_2$ loss. The rates of these two reactions increases with decreasing temperature, although the rate of $OH + NO + M$ is also pressure dependent. However, further modelling tests using the lowered $LHO_x$ production with the same $LNO_x$ as was measured at 360 hPa demonstrate even at 200 hPa and 220 K, the reactions $OH + NO + M$ and $HO_2 + NO$ still accounts for more than 50% of the OH loss and 80% of the $HO_2$ loss, respectively. Thus, based on this information, we also expect the same subsequent $HO_x$-$NO_x$ chemistry to occur in the upper troposphere as shown for the pressures and temperatures here."

We have also made the following addition to the first paragraph of the Conclusions at Line 288: "It took 3% NO and 3% HONO to resolve the measured-modelled discrepancy in these laboratory experiments, where the sparks occurred in a flow tube with laminar flow and a fast air velocity. In the atmosphere, the percentage of NO or HONO reacting with $LHO_x$ could be lower or higher than 3%, depending on the turbulence and air velocity where the lightning flash occurs, and likely varies from one lightning flash to the next. But the overall conclusion, that the $HO_x$ generated outside the hot channel only partially interacts with the hot channel products, will still be true in the atmosphere.

Additionally, these results indicate only..."

*J. B. Burkholder, S. P. Sander, J. Abbatt, J. R. Barker, C. Cappa, J. D. Crounse, T. S. Dibble, R. E. Huie, C. E. Kolb, M. J. Kurylo, V. L. Orkin, C. J. Percival, D. M. Wilmouth, and P. H. Wine "Chemical Kinetics and Photochemical Data for Use in Atmospheric Studies, Evaluation No. 19," JPL Publication 19-5, Jet Propulsion Laboratory, Pasadena, 2019 http://jpldataeval.jpl.nasa.gov.

**Line 108: Could your detection limit (20ppbv?) for O3 be too low to capture it?**

Yes, it is entirely possible some $O_3$ is formed that is below our detection limit. However, we do not have an $O_3$ analyzer with a lower detection limit that can also make measurements at different pressures and at a rate of 1 Hz or faster.

**If you generate short wave UV radiation from the sparks, O3 should probably be formed both from O2 and from NO2. The amount of O3 generated would be dependent on the amount of NO added and can also react with OH and HO2. Is the timescale of these reactions not relevant to the decay of HOx or can you exclude the impact of O3 via the model simulations?**

Even if $O_3$ is formed, the reactions of $O_3$ with OH and $HO_2$ are not very fast relative to the timescales we are considering here. From the most recent JPL evaluation:

$k_{OH+O_3}$= 7.3×10$^{-14}$ cm$^3$ molecules$^{-1}$ s$^{-1}$

$k_{HO_2+O_3}$= 1.9×10$^{-15}$ cm$^3$ molecules$^{-1}$ s$^{-1}$

If we run a version of the 3% NO + 3% HONO model runs that also includes 200 ppbv of $O_3$ (an amount we surely would have detected), we can see the effect of $O_3$ on the OH and $HO_2$ decays is not very substantial, especially for OH:

[Figure]

Based on this information, we feel justified in neglecting any $O_3$ possibly generated below our detection limit.

A new sentence has been added at Line 142: "Model tests confirmed that even if up to 20 ppbv of $O_3$ (our limit of detection) had been generated in the laboratory experiments, it would not have significantly affected the $HO_x$ decays, so $O_3$ was not included in any of the model runs shown here."

**Lines 114 f.: Could the losses for HOx and NO be different, e.g. through wall effects?**

Wall loss rates for OH and $HO_2$ are about 0.9 s$^{-1}$ and 0 s$^{-1}$, respectively (from Line 142), and there is not enough time for NO and $NO_2$ to have any wall interactions. It is

possible that there are differences between the $HO_x$ and $NO_x$ losses, but given that we cannot measure the $HO_x$ diffusion directly, assuming $HO_x$ and $NO_x$ diffuse similarly because of their similar diffusion coefficients is the best we can do.

**Line 119: Did you also change the location of the sampling or are you referring to the different locations of the discharge generation?**

The location of the sampling was fixed; this line refers to the different locations for the discharge. We have rewritten the end of the sentence from "…flow tube positions" to "…discharge positions in the flow tube" for clarity.

**Line 137: Do these cases also include 0ppbv of added NO, only considering spark NOx?**

Yes. These cases are the dotted green lines shown in subplots A and B on Figures 2, S7, S8, and S9.

**Line 149: Which reactions do you expect to occur? NO + HO2 -> NO2 + OH, NO + OH -> HONO, NO2 + OH -> HNO3? Which one is dominant?**

OH + NO $\rightarrow$ HONO and $HO_2$ + NO $\rightarrow$ OH + $NO_2$ are by far the most dominant reactions at all pressures, according to the modelling. The next most important reaction for $HO_2$ is OH + $HO_2$ $\rightarrow$ $H_2O$ + $O_2$ and for OH is OH + HONO $\rightarrow$ $H_2O$ + $NO_2$ (for the model runs that include 3% NO + 3% HONO). OH + $NO_2$ is at most the fourth most important OH loss, but is usually ranked even lower, while $HO_2$ + $NO_2$ is the third or fourth most important $HO_2$ loss.

**How important is recycling of OH through NO + HO2?**

This reaction is the only significant OH production pathway in the system (again, per the model runs). For the 0 ppbv of added NO cases this production pathway is always less than the total OH loss rate. For some of the cases with more than 0 ppbv of added NO, this recycling can be competitive with the total OH loss rate, at least initially.

**I recommend adding the reactions already in the introduction to explain the effects of NOx on HOx.**

We have added some discussion of the $HO_x$-$NO_x$ chemistry and which reactions dominate in a paragraph in the Conclusion (see response to an earlier question). Our overall goal with these experiments as mentioned in the Introduction was to determine whether $LHO_x$ was reacting with the spark or added NO, which is a not a question that these reactions can answer. The reaction OH + NO + M is no different whether the NO

comes from the spark or is added to the background air. So we will leave out discussing these reactions in the Introduction, and let the discussion in the Conclusion cover the important points.

**Figure 1: Are the differences between 0ppbv and 50ppbv of added NO really significant for 970hPa?**

Yes. The slopes and uncertainties for the individual experiments with 0 ppbv and 50 ppbv of added NO at 970 hPa are shown below (units are ln(molecules cm$^{-3}$) s$^{-1}$:

| OH slopes | Experiment 1 | Experiment 2 |
|---|---|---|
| 0 ppbv added NO | -15±1 | -14.0±0.8 |
| 50 ppbv added NO | -21±1 | -18±1 |
| HO$_2$ Slopes | | |
| 0 ppbv added NO | -14±1 | -11.2±0.9 |
| 50 ppbv added NO | -22±2 | -27±1 |

We have added a plot showing the slopes and uncertainties for all the different NO addition experiments at all four pressures to the Supplement as Figure S5. The sentence at Line 146 has been modified as follows (modification underlined): "As an increasing amount of NO was added to the air flow in the laboratory experiments, the OH and HO$_2$ decays became progressively steeper, as shown Figure 1 (970 hPa and 360 hPa), Figure S4 (770 hPa and 570 hPa), and Figure S5 (average slopes for all experiments)."

[Figure]

**Figure S5.** Average slopes for the OH (A) and $HO_2$ (B) decays from the NO addition experiments at different pressures.

**Line 155: How many measurements are included in each data point?**

Each OH or $HO_2$ data point contains data from 1-2 experimental decays (Line 152), and within each experiment, 4 sparks were made at each position in the flow tube (newly added Line 101). Because the GTHOS laser is set off the OH absorption wavelength for one of these four sparks (see replies to Reviewer 1), this means each OH or $HO_2$ data point contains 3 or 6 measurements (for 1 or 2 decays, respectively).

The Figures 1 and S4 captions have been modified as follows (modifications are underlined): "Laboratory decays of OH (A,B), $HO_2$ (C,D), and net $HO_x$ (E,F) at 970 hPa (A,C,E) and 360 hPa (B,D,F). The markers are the averaged data points containing 3 or 6 measurements from 1 or 2 laboratory experiments, respectively. The markers at time zero are the averaged extrapolated values from the decays. The lines on A, B, C, D are the linear fits to the individual decays. Error bars are the standard deviation from averaging the multiple laboratory measurements."

**Line 166 ff.: Do I understand correctly that in your experiment, HOx and NOx is both generated from the spark?**

Yes, both $HO_x$ and $NO_x$ are generated in the sparks, but generating both is not unique to our experiments. Anytime a spark occurs in air with water vapor it will generate both $HO_x$ and $NO_x$.

**So, for the 0% spark NOx case in your model, do you assume an initial HOx mixing ratio based on the experiments? Please clarify.**

Yes, except for the model run where we tested for HONO production, all the model runs use the same initial OH and $HO_2$ that was determined from the laboratory experiments, including the 0% spark $NO_x$ case, the 100% spark $NO_x$ case, and the cases using a small percentage of spark $NO_x$. For example, on Figure 2, for each subplot, the 0% spark $NO_x$ model case, the 100% spark NOx model case, and the laboratory data all start with the same initial $HO_x$. We explain the choice of initial $HO_x$ at Line 137 manuscript: "The initial OH and $HO_2$ determined from the extrapolation of the laboratory decays, scaled down 10-fold, were chosen as the initial OH and $HO_2$ (respectively) for the model runs." We have rewritten the next sentence to help clarify that the same initial $HO_x$ was used for all the runs using different $NO_x$, from "For $NO_x$, three cases were tested. " to "Using this same initial $HO_x$, three cases using different amounts of initial $NO_x$ were tested."

**Figure 2: Is the scaling of the y axis for panels C-F the same as for A-B? The minimum is not visible. And does the OH axis somehow relate to the HO2 axis? - The outline of the figure is a bit irritating (2 black outlines for the left and 3 for the right panels).**

We have updated Figure 2 to properly show the scaling on all the subplots. We have also changed the default outlining to completely go around all four sides on every subplot on this figure and all the other figures as well. The OH axis and $HO_2$ axis for the same NO addition experiment have the same scaling, so they are related that way.

[Figure]

**Figure 2:** Comparison of measured OH (A,C,E) and $HO_2$ (B,D,F) laboratory decays and two model decays at 770 hPa and (A,B) 0 ppbv of added NO, (C,D) 50 ppbv of added NO, and (E,F) 100 ppbv of added NO. The dashed purple lines are the model decay with only the added NO, and includes no $NO_x$ from the spark, and the dotted green lines are the model decay with the added NO and all of the spark $NO_x$. The blue circles are the average laboratory measurements and average extrapolated value at time zero, while the dashed-dotted blue lines are the individual

extrapolated linear fits to the laboratory data. Error bars are the standard deviation from averaging multiple measurements.

**Line 186: Does this mean that the default model run only includes HOx from the spark?**

We think that by "default model runs" you mean the model runs that produce the purple dashed lines. They are initiated with the measured $HO_x$, no $LNO_x$ from the spark, and the amount of NO added upstream in the flow tube that is indicated on Figures 2, S7, S8, and S9. So for the experiments with 0 ppbv of added NO, these model runs only include $HO_x$, 20 ppbv of CO, and OH wall loss.

**Or NOx from the spark is increased by x% because the amount of LNOx is uncertain?**

No, we do not add more $NO_x$ on top of the spark $NO_x$ and added NO. The model runs that give the OH and $HO_2$ decays shown by the green dashed line are initiated by the measured $HO_x$, the measured $LNO_x$ produced only by the sparks, and the amounts of added NO shown in the figure.

**Does this mean you were previously able to reproduce the HOx decay in the model without assuming spatial separation?**

No, we were never able to reproduce the $HO_x$ decays in a model without assuming that $HO_x$ was reacting with much less than 100% of the spark $NO_x$.

In this line, we are referring to the modelling we did in a previous study, Jenkins et al., 2021, which was very similar to the modelling we are doing here (e.g. with the 3 different modelling cases containing 100%, 0%, and a small % of spark $NO_x$). We also briefly talked about this study in the second paragraph of the Introduction. Much like how we needed to include ~3% of the spark $NO_x$ to help resolve the measure-modelled discrepancy in this study, in that previous study we used 0.5% of the spark $NO_x$ to reduce the discrepancy. The subsequent sentences explain why that 0.5% was not enough to resolve the discrepancy in this study.

We have rewritten this Line to clarify this information: "When we first observed this measured-modelled discrepancy in Jenkins et al., 2021, we were able to resolve the discrepancy for both OH and $HO_2$ by including just 0.5% of the spark $NO_x$ in a model run."

**Lines 249 ff.: How about upper tropospheric pressures? Would we expect a lower agreement given that the pressure is even lower?**

Possibly but we would not read too much into this lower agreement at this point. In order to conduct experiments at lower pressures we increase the air velocity in the flow tube, leading to measurements made over shorter and shorter timeframes. This apparent disagreement may be therefore partly driven by the experimental setup. Additionally, as we explain in the subsequent paragraph there is a lot of chemistry happening very rapidly when the discharge first occurs, and we do not have a model that can reproduce any of those processes. It may be that because we are making our first measurement faster at 360 hPa than the other pressures, we are catching some of that initial spark chemistry that we cannot fully model. Thus, this disagreement between the model and laboratory may be due more to our own laboratory or model limitations than something that is physically meaningful in the atmosphere.

**Table 1: What are the units of these values?**

Units (all ppbv) have been added to the table.

**Line 258: Are you able to measure HONO and confirm the model results?**

Unfortunately, we do not have an instrument able to measure HONO. If we did, we absolutely would have conducted experiments with it hooked up to the laboratory setup to test the model results.

A few other minor corrections/clarifications were made in the text (modifications have been underlined):

Line 40: "At the center of a lightning flash is a ~1-2 cm diameter core (Rakov and Uman, 2006) with air temperatures exceeding 30,000K (Orville, 1968a)."

Line 93: "To capture the $LHO_x$ decay, the copper rods were moved by a driver system so discharges were generated in 5 different positions in the flow tube…"

Line 215: Incorrect name spelling in reference; "Brandbold" corrected to "Brandvold".

Line 294: "and it is likely that UV is also responsible…" has been changed to "…and UV may also be responsible…"

Second sentence of the Abstract: "…which could rapidly remove…" has been changed to "which would rapidly remove…"